# Toward a Well-Calibrated Discrimination via Survival Outcome-Aware Contrastive Learning

**Dongjoon Lee**[*]
Chung-Ang University
dongzza97@cau.ac.kr

**Hyeryn Park**[*]
Chung-Ang University
hyeryn2000@cau.ac.kr

**Changhee Lee**
Korea University
changheelee@korea.ac.kr

## Abstract

Previous deep learning approaches for survival analysis have primarily relied on ranking losses to improve discrimination performance, which often comes at the expense of calibration performance. To address such an issue, we propose a novel contrastive learning approach specifically designed to enhance discrimination *without* sacrificing calibration. Our method employs weighted sampling within a contrastive learning framework, assigning lower penalties to samples with similar survival outcomes. This aligns well with the assumption that patients with similar event times share similar clinical statuses. Consequently, when augmented with the commonly used negative log-likelihood loss, our approach significantly improves discrimination performance without directly manipulating the model outputs, thereby achieving better calibration. Experiments on multiple real-world clinical datasets demonstrate that our method outperforms state-of-the-art deep survival models in both discrimination and calibration. Through comprehensive ablation studies, we further validate the effectiveness of our approach through quantitative and qualitative analyses.

## 1   Introduction

Deep learning models have gained significant attention in survival analysis (also known as time-to-event analysis), which focuses on predicting and understanding the occurrence of an adverse event (e.g., death) as a function of time. The utility of survival models is typically evaluated through two key aspects: discrimination and calibration. Discrimination measures a model's ability to differentiate between patients with varying risks, prioritizing those more likely to experience the event. Calibration, on the other hand, assesses how well a model's predictions align with the observed event distribution. In other words, a well-calibrated model offers predictions that closely match actual survival outcomes, offering crucial prognostic value to clinicians.

In an effort to enhance the discriminative power of survival models, ranking loss functions [1, 2, 3, 4] are frequently employed. These functions aim to maximize a relaxed proxy of the concordance index (C-index), a well-established metric for evaluating the quality of patient rankings based on the risk predictions of survival models [5]. However, a notable improvement in discriminative power often comes at the expense of calibration performance, which can negatively affect the clinical utility of predicted survival outcomes. For example, a poorly calibrated model that overestimates risk may lead to unnecessary treatment or testing for patients. Conversely, underestimating risk may result in patients with adverse conditions not receiving appropriate care.

Contrastive learning [6, 7, 8] is a framework that aims to learn an embedding space where similar samples are mapped to nearby locations, while dissimilar samples are pushed farther apart. Notably, although commonly used in an unsupervised fashion, this framework shares a fundamental principle

---

[*]Equal contribution

38th Conference on Neural Information Processing Systems (NeurIPS 2024).

with ranking losses: both focus on the relative ranking of samples for better discrimination. However, unlike ranking losses, which directly modify differences in model outputs, contrastive learning operates on the relative distances of representations in the embedding space. This distinction offers the potential to overcome misalignment issues caused by direct comparisons and modifications of model outputs inherent in the ranking-based approaches. To leverage this potential and further enhance survival models, we propose incorporating informed sampling into the contrastive learning framework. By assigning lower penalties to potential false negative pairs based on the similarity of their survival outcomes, we instill an inductive bias in the model, encouraging it to learn that samples with similar event times likely share similar clinical statuses. This allows us to achieve strong discrimination without directly comparing or manipulating model outputs, thus potentially improving calibration as well.

**Contribution.** In this paper, we propose a novel contrastive learning approach, specifically designed for survival analysis, to enhance both discrimination and calibration performance.[2] Motivated by the assumption that patients with similar event times share similar clinical status, we contrast samples in the latent space based on their survival outcomes. However, unlike standard contrastive methods, we leverage the model output solely for maximum likelihood estimation (MLE), thereby promoting better calibration. This is achieved by incorporating importance sampling into our contrastive learning framework, where we assign weights proportional to the differences in survival outcomes. This guides our model to learn appropriate similarity relationships between clinical features relevant to survival analysis. Through experiments on multiple real-world clinical datasets, we demonstrate the superiority of our method in both discrimination and calibration performance, outperforming state-of-the-art deep survival models. We further provide comprehensive quantitative and qualitative analyses through ablation studies to showcase the impact of our novel contrastive learning approach tailored for survival analysis.

## 2 Related Works

### 2.1 Deep Learning Approaches to Survival Analysis

Deep learning-based survival models have garnered significant attention due to their ability to provide non-parametric estimations of the underlying discrete-time survival distribution, particularly conditional hazard or survival functions. Negative log-likelihood (NLL) has been widely used in survival analysis to estimate various survival quantities, such as conditional hazard functions [9, 10, 11], probability mass functions of event times [1], or survival functions [4]. MLE, based on NLL, provides unbiased estimates of these quantities under the assumption of ignorable censoring, where the event and censoring are independent given the input features. Building on this foundation, recent deep learning-based survival models often incorporate auxiliary losses alongside the NLL loss to further enhance model performance.

**Ranking Loss.** Ranking loss, often augmented with the NLL loss, has been widely employed in recent deep survival models to enhance the discriminative power of survival predictions [1, 2, 3, 4]. These ranking loss functions, which are differentiable approximations or upper bounds of the negative C-index [12], are typically based on exponential [1, 2], log-sigmoid [4, 13], or linear [3] functions. In particular, these approaches directly utilize model outputs, such as conditional hazard or survival functions, to establish pairwise ordering of comparable individual risks for better discrimination. However, since ranking loss primarily focuses on samples with earlier event times (often ignoring censored samples), these models may struggle to capture the full distributional characteristics of time-to-event outcomes. This can adversely affect calibration performance, potentially leading to inaccurate predictions regarding the observed time-to-event outcomes [2].

**Calibration Loss.** In healthcare applications, while achieving high discriminative power through ranking loss is crucial, well-calibrated predictions are equally important for effective clinical decision-making. Notably, recent approaches [14, 2] have prioritized enhancing calibration by minimizing the rank probability score (RPS), a prediction error metric specifically tailored for survival analysis. Furthermore, the authors in [15] have introduced a novel approach that transforms D-Calibration [16] into a differentiable objective, calculating the squared difference between observed and predicted events across multiple time intervals. In contrast to these methods, our proposed approach does not

---

[2]Source code for ConSurv is available in https://github.com/dongzza97/ConSurv

explicitly incorporate a calibration loss function. Nevertheless, it achieves comparable or even superior calibration performance, demonstrating the effectiveness of our contrastive learning framework in implicitly preserving calibration alongside discrimination.

## 2.2 Contrastive Learning

Contrastive learning [6, 7, 17] is a framework that learns an embedding space that effectively discriminates among samples, by mapping positive pairs (similar samples) closer together, while pushing negative pairs (dissimilar samples) farther apart. While contrastive learning has made significant progress, our review of related work will focus on methods that explore various strategies for injecting inductive bias to enhance representation learning.

**Exploiting Continuous Labels.** Inspired by the success of incorporating label information into contrastive learning for classification tasks [8], recent research has extended this concept to regression tasks by exploiting continuous label information for constructing positive and negative pairs. Zha et al. [18] employ hard thresholding on label differences to generate positive and negative samples, capitalizing on the idea that samples with similar target values should be mapped closer in the latent space. Kerdabadi et al. [19] apply a similar approach in a dynamic time-to-event setting, defining positive pairs as those within a specific time window and weighting negative pairs proportionally to their time difference. However, this weighting scheme's heavy reliance on absolute time differences can destabilize the contrastive loss during training.

**Weighting Negative Samples.** Recent works have significantly improved contrastive learning by incorporating inductive bias in the selection of negative samples. This includes focusing more weights on hard negatives that challenge the model for more effective discrimination [20, 21, 22, 23] and avoiding potential false negatives that might mislead the model [24, 25]. Many of these studies address negative sample selection by employing informed sampling techniques to identify informative hard negatives or potential false negatives. Notably, Robinson et al. [20] leverage importance sampling by proposing a similarity-based distribution that prioritizes hard-to-distinguish negative samples.

# 3 Problem Formulation

## 3.1 Preliminary: Discrete-Time Survival Analysis

Suppose we are given a discrete-time survival dataset comprising $N$ patients, denoted as $\mathcal{D} = \{(\mathbf{x}_i, \tau_i, \delta_i)\}_{i=1}^N$. Each patient $i$ is represented by the input feature $\mathbf{x}_i \in \mathcal{X}$ where $\mathcal{X}$ is the input space, and the observed survival outcomes, $\tau_i \in \mathcal{T}$ and $\delta_i \in \{0, 1\}$. Here, $\tau_i$ and $\delta_i$ indicate the time elapsed until either an event of our interest (e.g., death) or censoring (e.g., lost to follow-up) occurs and whether the event was observed or not (i.e., right-censored), respectively. Throughout, we treat survival time as discrete and the time horizon as finite such that the set of possible survival times is defined as $\mathcal{T} = \{0, \ldots, T_{\max}\}$ with a pre-defined maximum time horizon $T_{\max}$.

The conditional hazard function, $\lambda : \mathcal{X} \times \mathcal{T} \to [0, 1]$, is the instantaneous risk of the event at time $t$ given feature $\mathbf{x}$ and is defined as $\lambda(t|\mathbf{x}) = \mathbb{P}(T = t|T \geq t, \mathbf{x})$. Then, we can represent the survival function $S : \mathcal{X} \times \mathcal{T} \to [0, 1]$ as follows:

$$S(t|\mathbf{x}) = \mathbb{P}(T > t|\mathbf{x}) = \prod_{t' \leq t} (1 - \lambda(t'|\mathbf{x})) \tag{1}$$

which is a non-increasing function of $t$, indicating the probability of the event occurring after time $t$ given feature $\mathbf{x}$. Equivalently, we could estimate the risk function, $R : \mathcal{X} \times \mathcal{T} \to [0, 1]$, which represents the probability of the event occurring at or before time $t$ given feature $\mathbf{x}$, i.e., $R(t|\mathbf{x}) = \mathbb{P}(T \leq t|\mathbf{x}) = 1 - S(t|\mathbf{x})$.

Then, we can achieve likelihood-based estimates for the hazard function, $\hat{\lambda}$, by minimizing the following negative NLL loss:

$$\mathcal{L}_{NLL} = -\sum_{i=1}^N \left[ \delta_i \log \hat{p}(\tau_i|\mathbf{x}_i) + (1 - \delta_i) \log \hat{S}(\tau_i|\mathbf{x}_i) \right] \tag{2}$$

where $\hat{p}(t|\mathbf{x}) = \hat{\lambda}(t|\mathbf{x})\hat{S}(t - 1|\mathbf{x})$ represents the estimate for the probability of an event occurring at time $t$, i.e., $\mathbb{P}(T = t|\mathbf{x})$. Here, (2) exploits two pieces of information from the survival data: (i)

when the event is observed (i.e., $\delta_i = 1$), we know that the event occurred at time $\tau_i$, and (ii) when the event is not observed (i.e., $\delta_i = 0$), we are aware that the event will occur after time $\tau_i$.

## 3.2 Ranking Loss for Survival Analysis

Ranking loss specifically targets enhancing the discriminative power of survival models, which is crucial for better distinguishing between patients based on their risk. Suppose we are given the risk function, $R$, that quantifies the risk of a given patient with $\mathbf{x}$ at any time $t$. We aim to penalize instances where the risks assigned to a pair of patients are incorrectly ordered (i.e., assigning a lower risk to patient $i$ who died before patient $j$). This can be achieved by the ranking loss, which is formally given as follows:

$$\mathcal{L}_{Rank} = \sum_{i \neq j} A_{i,j} \cdot \mathbb{I}\big(R(\tau_i|\mathbf{x}_i) < R(\tau_i|\mathbf{x}_j)\big) \approx \sum_{i \neq j} A_{i,j} \cdot \eta\big(R(\tau_i|\mathbf{x}_i), R(\tau_i|\mathbf{x}_j)\big), \qquad (3)$$

where $\eta$ is a function that relaxes the non-differentiable indicator function, $\mathbb{I}$. Here, $A_{i,j} = \mathbb{I}(\delta_i = 1, \tau_i < \tau_j)$ indicates acceptable pairs whose assigned risks are comparable.

Different deep survival models have employed various types of quantities for implementing the ranking loss. For example, Lee et al. [1] have applied the risk function, $R$, and set $\eta(x, y) = \exp(-(x-y)/\kappa)$, a convex function that both penalizes the wrongly ordered pairs and encourages the correctly ordered pairs. The same convex function has been employed utilizing the survival function, $S$ [2]. Similarly, Steck et al. [13] utilizes $\eta(x, y) = \log \sigma(y - x)$, which is a lower bound of the C-index, with the risk function, $R$. Chi et al. [3] employ the hazard function $h$ with a linear ranking function $\eta(x, y) = -(x - y)$.

**Challenges.** While combining NLL with ranking loss shows a notable improvement in discriminative power, it often comes at the expense of calibration performance, potentially harming the clinical utility of predicted survival outcomes. We suspect this is primarily due to how ranking loss directly modifies model outputs to order predicted risks, potentially leading to misalignment with the actual risk distribution. In this paper, we propose a different approach. Our method focuses on increasing discriminative power not by directly utilizing survival outcomes but by exploiting the embedding space through our novel contrastive learning framework. This approach preserves the calibration performance achieved by the NLL loss while enhancing discriminative capabilities.

# 4 Method

To address the challenges described above, we propose a novel **Con**trastive learning approach to a deep **Sur**vival model, which we refer to as **ConSurv**. The proposed method consists of three key components as illustrated in Figure 1:

- *Encoder* (parameterized by $\theta$), $f_\theta : \mathcal{X} \to \mathcal{H}$, takes features $\mathbf{x} \in \mathcal{X}$ as input and outputs latent representations, i.e., $\mathbf{h} = f_\theta(\mathbf{x})$.
- *Projection head* (parameterized by $\psi$), $g_\psi : \mathcal{H} \to \mathbb{R}^d$, maps latent representations $\mathbf{h}$ to the embedding space where contrastive learning is applied, i.e., $\mathbf{z} = g_\psi(\mathbf{h})$.
- *Hazard network* (parameterized by $\phi$), $f_\phi : \mathcal{H} \times \mathcal{T} \to [0, 1]$, predicts the hazard rate at each time point $t \in \mathcal{T}$ given the input latent representation $\mathbf{h}$, i.e., $\hat{\lambda}(t|\mathbf{x}) = f_\phi(\mathbf{h}, t) = f_\phi(f_\theta(\mathbf{x}), t)$.

Motivated by the core concept of contrastive learning frameworks, we aim to differentiate each sample from other *semantically different* samples based on their *survival outcomes*. This allows us to overcome the limitation of ranking loss, which arises from the direct comparison of model outcomes in the form of risk/survival functions. Instead, we contrast samples in the latent space based on their survival outcomes and utilize the model outcome solely for the NLL loss to encourage better calibration.

## 4.1 Contrastive Learning for Survival Analysis

Our novel contrastive learning framework imposes lower penalties on potential false negative samples based on the similarities in their survival outcomes. More specifically, given an anchor sample, we define potential false negatives as samples with a small difference in the corresponding time-to-event.

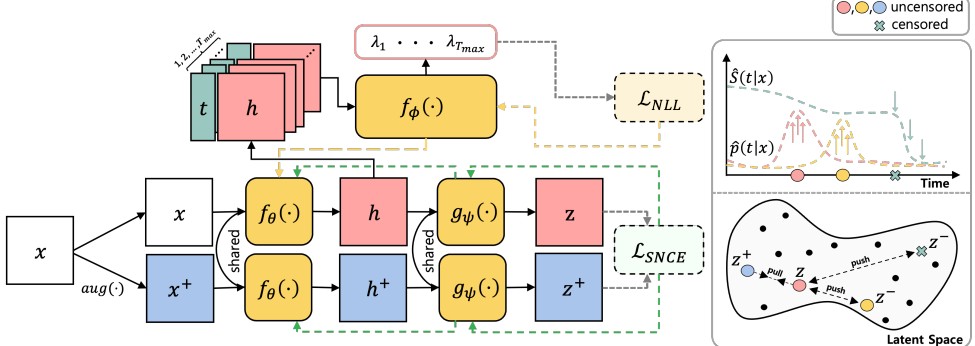

Figure 1: An illustration of the network architecture for ConSurv.

This inherently aligns with our inductive bias that patients with similar survival outcomes should share similar clinical status, which manifests through similar representations.

For each anchor sample $\mathbf{x} \sim p_X$, the noise-contrastive estimation (NCE) objective aims to learn mapping $f = g_\psi \circ f_\theta$ utilizing a positive sample with the same semantic meaning, i.e., $\mathbf{x}^+ \sim p_+$, and negative samples with (supposedly) different semantic meanings, i.e., $\mathbf{x}^- \sim q$, as follows [20]:

$$\mathbb{E}_{\substack{\mathbf{x} \sim p_X \\ x^+ \sim p_{X^+}}} \left[ -\log \frac{e^{s(\mathbf{x},\mathbf{x}^+)}}{M \cdot \mathbb{E}_{\mathbf{x}^- \sim q}[e^{s(\mathbf{x},\mathbf{x}^-)}]} \right] \tag{4}$$

where $M$ is the scaling term which is set to the batch size and $s : \mathcal{X} \times \mathcal{X} \to [-1,1]$ is the similarity score between two samples. Here, we use cosine similarity in the embedding space, i.e., $s(\mathbf{x},\mathbf{x}') = \frac{f(\mathbf{x})^\top f(\mathbf{x}')}{\|f(\mathbf{x})\|\|f(\mathbf{x}')\|}$. For notational convenience, we omit the corresponding temperature $\nu$ and write $e^{s(\mathbf{x},\mathbf{x}^+)}$ to denote $e^{s(\mathbf{x},\mathbf{x}^+)/\nu}$.

The key aspect in (4) for obtaining embeddings with discriminative power lies in how we select negative samples that the anchor sample should be distinguished from. To reflect the differences in the time-to-events in the embedding space, we design a novel negative distribution, $q$, by utilizing the available information from survival outcomes.

**Importance Sampling using Survival Outcomes.** To accurately distinguish patients based on their time-to-event outcomes, we fully utilize the time-to-event information for designing $q$ based on the following inductive bias: similar patients are more likely to experience the event at similar time points than the ones who are not. Hence, given an anchor $(\mathbf{x}, \tau)$ and a negative sample $(\mathbf{x}^-, \tau^-)$, we define the weight function as follows:

$$w(\tau^-; \tau) = 1 - e^{-|\tau - \tau^-| / \sigma} \tag{5}$$

where $\sigma > 0$ is a temperature coefficient. From this point forward, we will slightly abuse the notation and write $w(\mathbf{x}^-; \mathbf{x})$ to denote $w(\tau^-; \tau)$ to be consistent with the notation used for the negative distribution. This function is a variant of the Laplacian kernel, where the weight increases as the time difference increases. That is, we assign larger weights for samples with large differences in the time-to-events, and smaller weights for samples with small time differences. Utilizing the weight function in (5), we can define the negative distribution, $q$, as:

$$q(\mathbf{x}^-; \mathbf{x}) = \frac{1}{Z} w(\mathbf{x}^-; \mathbf{x}) p(\mathbf{x}^-), \tag{6}$$

where $Z$ is the normalizing constant. Then, using the importance sampling technique, we can approximate the expectation of the similarity score of negative samples drawn from the negative distribution in (6) as:

$$\mathbb{E}_{\mathbf{x}^- \sim q}\left[ e^{s(\mathbf{x},\mathbf{x}^-)} \right] = \mathbb{E}_{\mathbf{x}^- \sim p}\left[ \frac{q(\mathbf{x}^-; \mathbf{x})}{p(\mathbf{x}^-)} \cdot e^{s(\mathbf{x},\mathbf{x}^-)} \right] \approx \frac{1}{Z \cdot M} \sum_{j=1}^{M} w(\mathbf{x}_j^-; \mathbf{x}) \cdot e^{s(\mathbf{x},\mathbf{x}_j^-)} \tag{7}$$

where the normalizing constant for the empirical distribution can be given as $Z = \frac{1}{M}\sum_{j=1}^{M} w(\mathbf{x}_j^-; \mathbf{x})$. Overall, the survival outcome-aware NCE (SNCE) loss in (8), mitigating the effect of potential false negatives that have similar survival outcomes, can be given as the following:

$$\mathcal{L}_{SNCE} = \sum_{i=1}^{N}\left[-\log \frac{e^{s(\mathbf{x}_i,\mathbf{x}_i^+)}}{\frac{1}{Z}\sum_{j=1}^{M} w(\mathbf{x}_j^-; \mathbf{x}_i) \cdot e^{s(\mathbf{x}_i,\mathbf{x}_j^-)}}\right]. \tag{8}$$

## 4.2 Handling Right-Censoring

However, in a survival dataset, there exist samples that are right-censored, offering partial information about the corresponding survival outcomes which indicates that the event will occur sometime after the censoring time. Consequently, not every pair in the survival dataset has comparable survival outcomes. Specifically, there are three different cases of sample pairs to consider:

- **Case 1:** Both samples are uncensored (i.e., have observed events). We can directly compare the two time-to-events of any given pair.
- **Case 2:** One is uncensored and the other is censored. Similar to the acceptable pairs in ranking loss, we can only compare a pair when sample $i$ experiences an event while sample $j$ is censored after that event time (implying no event has occurred by that time), i.e., $\tau_i < \tau_j$.
- **Case 3:** Both samples are censored. No comparison is possible as we do not know when the actual events have occurred for both samples.

Considering these cases, we redefine the weight function considering the right-censoring as $\tilde{w}(\tau_j; \tau_i) = I_{i,j} \cdot w(\tau_j; \tau_i)$ where $I_{i,j}$ indicates the comparable cases as follows:

$$I_{i,j} = \begin{cases} 1 & \text{if } (\delta_i = 1, \delta_j = 1) \vee ((\delta_i = 1, \delta_j = 0) \wedge (\tau_i < \tau_j) \wedge (|\tau_i - \tau_j| \geq \alpha)) \\ 0 & \text{otherwise} \end{cases}$$

where $\alpha$ is a hyperparameter that represents a margin to ensure a minimum time difference. Please see Appendix C.3 for more details about the effect of $\alpha$.

## 4.3 Network Description

In this subsection, we provide a detailed network description of ConSurv and outline the learning procedure using a discrete-time survival dataset, $\mathcal{D} = \{(\mathbf{x}_i, \tau_i, \delta_i)\}_{i=1}^{N}$.

**Negative Log-likelihood.** Based on the encoder and the hazard network, we can provide the estimated hazard function given the input feature $\mathbf{x}$ at each time point $t \in \mathcal{T}$ as $\hat{\lambda}(t|\mathbf{x}) = f_\phi(f_\theta(\mathbf{x}), t)$. As our hazard estimate is defined as a function of time given an input feature, we can naturally model the time-varying effect of input features (thus, more complex relations) on risk/survival functions. Then, we can compute the NLL loss, $\mathcal{L}_{NLL}^{\theta,\phi}$, in (2) by plugging in $f_\phi(f_\theta(\mathbf{x}), t)$ into $\hat{p}$ and $\hat{S}$ as follows:

$$\hat{p}(\tau|\mathbf{x}) = f_\phi(f_\theta(\mathbf{x}), \tau)\prod_{t' \leq \tau-1}\left(1 - f_\phi(f_\theta(\mathbf{x}), t')\right), \quad \hat{S}(\tau|\mathbf{x}) = \prod_{t' \leq \tau}\left(1 - f_\phi(f_\theta(\mathbf{x}), t')\right). \tag{9}$$

**Contrastive Learning.** To train the proposed method using our contrastive loss introduced in (8), we construct augmented samples based on the state-of-the-art contrastive learning framework specifically designed for tabular data [26]. For each sample in a batch of $M$ samples, we construct a corrupted version of the original sample following the marginal corruption process. So, given $\mathbf{x}_i$ as an anchor, we set the corrupted sample, i.e., $\tilde{\mathbf{x}}_i$, as positive and all the other corrupted samples, i.e., $\tilde{\mathbf{x}}_j$ for $j \neq i$, as negative. By passing the original, positive, and negative samples through $f = g_\psi \circ f_\theta$, we can compute our survival outcome-based contrastive learning loss function as defined in (8).

Overall, we can estimate the hazard function by training ConSurv with a loss function that combines the NLL loss and the SNCE loss as the following:

$$\mathcal{L}_{Total}^{\theta,\phi,\psi} = \mathcal{L}_{NLL}^{\theta,\phi} + \beta\mathcal{L}_{SNCE}^{\theta,\psi}, \tag{10}$$

where $\beta$ is a balancing coefficient that trade-offs between the two loss terms; the effect of $\beta$ is provided in C.3. Please find the pseudo-code of ConSurv in H.

Table 1: Discrimination and calibration of survival models: mean and standard deviation values for CI, IBS, and DDC, along with the number of successful D-calibration tests.[3]

| | METABRIC | | | | NWTCO | | | |
|---|---|---|---|---|---|---|---|---|
| **METHOD** | CI ↑ | IBS ↓ | DDC ↓ | D-CAL | CI ↑ | IBS ↓ | DDC ↓ | D-CAL |
| CoxPH | $0.662_{\pm 0.015}$ | $\mathbf{0.170_{\pm 0.013}}$ | $0.108_{\pm 0.009}$ | **10** | $0.712_{\pm 0.031}$ | $0.101_{\pm 0.010}$ | $0.567_{\pm 0.036}$ | **10** |
| DeepSurv | $0.645_{\pm 0.014}$ | $0.188_{\pm 0.019}$ | $0.160_{\pm 0.051}$ | **10** | $0.645_{\pm 0.087}$ | $\mathbf{0.090_{\pm 0.017}}$ | $0.642_{\pm 0.087}$ | 9 |
| DeepHit | $0.636_{\pm 0.039}$ | $0.205_{\pm 0.017}$ | $0.289_{\pm 0.004}$ | 0 | $0.713_{\pm 0.009}$ | $0.140_{\pm 0.048}$ | $0.616_{\pm 0.030}$ | 4 |
| DRSA | $0.635_{\pm 0.017}$ | $0.260_{\pm 0.036}$ | $0.199_{\pm 0.027}$ | 0 | $0.700_{\pm 0.037}$ | $0.272_{\pm 0.050}$ | $0.295_{\pm 0.127}$ | 0 |
| DCS | $0.598_{\pm 0.015}$ | $0.243_{\pm 0.022}$ | $0.090_{\pm 0.050}$ | 0 | $0.677_{\pm 0.031}$ | $0.098_{\pm 0.015}$ | $0.173_{\pm 0.051}$ | 6 |
| X-CAL | $0.658_{\pm 0.009}$ | $0.189_{\pm 0.018}$ | $\mathbf{0.068_{\pm 0.041}}$ | 0 | $0.677_{\pm 0.038}$ | $0.146_{\pm 0.023}$ | $\mathbf{0.133_{\pm 0.074}}$ | 4 |
| $\mathcal{L}_{NLL}$ | $0.608_{\pm 0.046}$ | $0.244_{\pm 0.047}$ | $0.086_{\pm 0.055}$ | 5 | $0.711_{\pm 0.051}$ | $0.102_{\pm 0.006}$ | $0.464_{\pm 0.049}$ | 10 |
| $\mathcal{L}_{NLL}$ & $\mathcal{L}_{NCE}$ | $0.660_{\pm 0.002}$ | $0.191_{\pm 0.014}$ | $0.115_{\pm 0.031}$ | 8 | $0.716_{\pm 0.048}$ | $0.100_{\pm 0.009}$ | $0.289_{\pm 0.012}$ | 8 |
| $\mathcal{L}_{NLL}$ & $\mathcal{L}_{Rank}$ | $0.643_{\pm 0.010}$ | $0.267_{\pm 0.025}$ | $0.155_{\pm 0.050}$ | 0 | $0.723_{\pm 0.042}$ | $0.210_{\pm 0.026}$ | $0.551_{\pm 0.037}$ | 0 |
| **CONSURV** | $\mathbf{0.688_{\pm 0.017}}$ | $0.183_{\pm 0.019}$ | $0.097_{\pm 0.025}$ | 6 | $\mathbf{0.729_{\pm 0.052}}$ | $\mathbf{0.090_{\pm 0.011}}$ | $0.222_{\pm 0.029}$ | **10** |

| | GBSG | | | | FLCHAIN | | | |
|---|---|---|---|---|---|---|---|---|
| **METHOD** | CI ↑ | IBS ↓ | DDC ↓ | D-CAL | CI ↑ | IBS ↓ | DDC ↓ | D-CAL |
| CoxPH | $0.677_{\pm 0.010}$ | $0.177_{\pm 0.003}$ | $0.220_{\pm 0.037}$ | **10** | $0.800_{\pm 0.009}$ | $0.100_{\pm 0.006}$ | $0.302_{\pm 0.016}$ | **10** |
| DeepSurv | $0.674_{\pm 0.006}$ | $0.179_{\pm 0.003}$ | $0.203_{\pm 0.024}$ | **10** | $0.798_{\pm 0.010}$ | $0.084_{\pm 0.011}$ | $0.294_{\pm 0.010}$ | **10** |
| DeepHit | $0.642_{\pm 0.058}$ | $0.207_{\pm 0.003}$ | $0.354_{\pm 0.052}$ | 1 | $0.798_{\pm 0.009}$ | $0.168_{\pm 0.003}$ | $0.500_{\pm 0.004}$ | 0 |
| DRSA | $0.679_{\pm 0.002}$ | $0.283_{\pm 0.014}$ | $0.402_{\pm 0.083}$ | 0 | $0.779_{\pm 0.004}$ | $0.226_{\pm 0.025}$ | $0.318_{\pm 0.044}$ | 0 |
| DCS | $0.693_{\pm 0.006}$ | $0.180_{\pm 0.009}$ | $0.166_{\pm 0.010}$ | 5 | $0.786_{\pm 0.012}$ | $0.105_{\pm 0.008}$ | $\mathbf{0.251_{\pm 0.050}}$ | 0 |
| X-CAL | $0.687_{\pm 0.006}$ | $0.179_{\pm 0.002}$ | $\mathbf{0.087_{\pm 0.018}}$ | 3 | $0.791_{\pm 0.006}$ | $0.110_{\pm 0.010}$ | $0.302_{\pm 0.097}$ | 3 |
| $\mathcal{L}_{NLL}$ | $0.675_{\pm 0.018}$ | $0.174_{\pm 0.002}$ | $0.126_{\pm 0.025}$ | 0 | $0.794_{\pm 0.017}$ | $0.104_{\pm 0.005}$ | $0.273_{\pm 0.015}$ | 9 |
| $\mathcal{L}_{NLL}$ & $\mathcal{L}_{NCE}$ | $0.687_{\pm 0.008}$ | $0.184_{\pm 0.003}$ | $0.193_{\pm 0.006}$ | 0 | $0.797_{\pm 0.012}$ | $0.111_{\pm 0.011}$ | $0.302_{\pm 0.043}$ | 8 |
| $\mathcal{L}_{NLL}$ & $\mathcal{L}_{Rank}$ | $0.687_{\pm 0.012}$ | $0.342_{\pm 0.034}$ | $0.359_{\pm 0.117}$ | 0 | $0.796_{\pm 0.014}$ | $0.184_{\pm 0.016}$ | $0.334_{\pm 0.056}$ | 0 |
| **CONSURV** | $\mathbf{0.696_{\pm 0.006}}$ | $\mathbf{0.175_{\pm 0.002}}$ | $0.180_{\pm 0.017}$ | 0 | $\mathbf{0.810_{\pm 0.011}}$ | $\mathbf{0.072_{\pm 0.057}}$ | $0.291_{\pm 0.055}$ | 9 |

| | SUPPORT | | | | SEER | | | |
|---|---|---|---|---|---|---|---|---|
| **METHOD** | CI ↑ | IBS ↓ | DDC ↓ | D-CAL | CI ↑ | IBS ↓ | DDC ↓ | D-CAL |
| CoxPH | $0.605_{\pm 0.006}$ | $0.196_{\pm 0.006}$ | $0.262_{\pm 0.009}$ | 0 | $0.857_{\pm 0.021}$ | $0.009_{\pm 0.001}$ | $0.966_{\pm 0.004}$ | **10** |
| DeepSurv | $0.599_{\pm 0.011}$ | $0.196_{\pm 0.007}$ | $0.258_{\pm 0.033}$ | 0 | $0.760_{\pm 0.037}$ | $0.010_{\pm 0.001}$ | $1.000_{\pm 0.000}$ | **10** |
| DeepHit | $0.502_{\pm 0.007}$ | $0.272_{\pm 0.003}$ | $0.336_{\pm 0.005}$ | 0 | $0.843_{\pm 0.021}$ | $0.020_{\pm 0.001}$ | $0.825_{\pm 0.002}$ | 0 |
| DRSA | $0.573_{\pm 0.007}$ | $0.263_{\pm 0.019}$ | $0.285_{\pm 0.097}$ | 0 | $0.810_{\pm 0.113}$ | $0.023_{\pm 0.022}$ | $\mathbf{0.650_{\pm 0.174}}$ | 0 |
| DCS | $0.597_{\pm 0.009}$ | $0.211_{\pm 0.012}$ | $0.169_{\pm 0.043}$ | 0 | $0.860_{\pm 0.022}$ | $0.010_{\pm 0.001}$ | $0.903_{\pm 0.037}$ | 9 |
| X-CAL | $0.604_{\pm 0.006}$ | $0.207_{\pm 0.013}$ | $0.182_{\pm 0.029}$ | 0 | $0.844_{\pm 0.033}$ | $0.011_{\pm 0.006}$ | $0.907_{\pm 0.029}$ | 9 |
| $\mathcal{L}_{NLL}$ | $0.607_{\pm 0.005}$ | $0.196_{\pm 0.008}$ | $0.122_{\pm 0.014}$ | 0 | $0.853_{\pm 0.017}$ | $0.009_{\pm 0.001}$ | $0.962_{\pm 0.008}$ | 10 |
| $\mathcal{L}_{NLL}$ & $\mathcal{L}_{NCE}$ | $0.609_{\pm 0.006}$ | $0.193_{\pm 0.006}$ | $0.130_{\pm 0.025}$ | 0 | $0.855_{\pm 0.018}$ | $0.009_{\pm 0.001}$ | $0.965_{\pm 0.006}$ | 10 |
| $\mathcal{L}_{NLL}$ & $\mathcal{L}_{Rank}$ | $0.610_{\pm 0.005}$ | $0.294_{\pm 0.007}$ | $0.232_{\pm 0.015}$ | 0 | $0.863_{\pm 0.017}$ | $0.120_{\pm 0.003}$ | $1.000_{\pm 0.000}$ | 0 |
| **CONSURV** | $\mathbf{0.618_{\pm 0.006}}$ | $\mathbf{0.192_{\pm 0.005}}$ | $\mathbf{0.144_{\pm 0.021}}$ | 0 | $\mathbf{0.865_{\pm 0.014}}$ | $\mathbf{0.004_{\pm 0.003}}$ | $0.961_{\pm 0.007}$ | **10** |

## 5   Experiment

In this section, we evaluate the performance of ConSurv and multiple survival models using several real-world clinical datasets. Further details about all the experiments are available in Appendix D.

### 5.1   Experiment Setup

**Datasets.**  We compare our proposed method and the benchmarks with the following four commonly used real-world clinical datasets: *METABRIC*, *NWTCO*, *GBSG*, *FLCHAIN*, *SUPPORT*, and *SEER*. For detailed descriptions of these datasets, please refer to D.1.

**Benchmarks.**  We compare ConSurv with six survival models that were selected based on their respective loss functions, which are critical for understanding trends in survival analysis performance. The evaluated models are based on i) partial log-likelihood including *CoxPH* [27] and *DeepSurv* [28], ii) ranking loss including *DeepHit* [1] and *DRSA* [10], and iii) calibration loss including *DCS* [2] and *X-CAL* [15]. Detailed methodological descriptions are provided in Appendix D.3.

**Performance Metric.** We evaluate the discriminative performance of ConSurv and benchmarks using the *integrated time-dependent C-index* (CI) [29] across all time points, where higher values indicate better performance. For calibration, we utilize the *integrated Brier score* (IBS) [30] and *distributional divergence for calibration* (DDC) [2], where lower values indicate better performance. Additionally, *D-Calibration* (D-CAL) [31] assesses calibration with results reported based on $p$-values; those exceeding 0.05 are noted as statistically significant. Comprehensive details on IBS, DDC, and D-CAL

---

[3]Table 1 presents the performance results obtained using 10 random seeds. For enhanced stability verification, we conducted additional experiments with 25 random seeds. Please refer to Table 10 in appendix C.5.

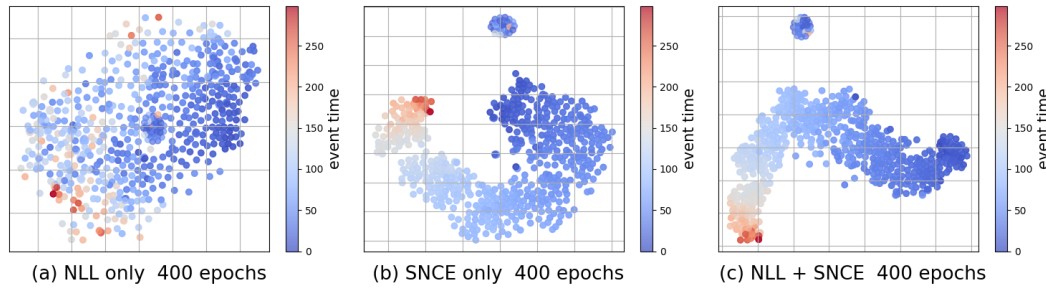

(a) NLL only  400 epochs     (b) SNCE only  400 epochs     (c) NLL + SNCE  400 epochs

Figure 2: t-SNE visualization for latent representations learned with $\mathcal{L}_{NLL}$ only, $\mathcal{L}_{SNCE}$ only, and ConSurvfor the METABRIC dataset, colored by event times (for uncensored samples).

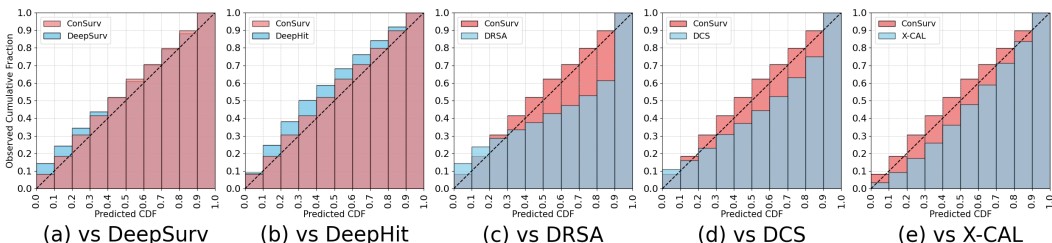

(a) vs DeepSurv     (b) vs DeepHit     (c) vs DRSA     (d) vs DCS     (e) vs X-CAL

Figure 3: Calibration plots for ConSurv in comparison with benchmarks for the METABRIC dataset.

are provided in Appendix D.2. In Appendix C.1, we further provide time-dependent C-index [5] and time-dependent Brier score [30] at different time points.

## 5.2 Quantitative Analysis Results

We report the CI, BS, DDC, and D-CAL of our proposed method and the benchmarks in Table 1. The results in Table 1 demonstrate that our method significantly outperforms all the benchmarks in discriminating among individual risks at the evaluated all time points, consistently across the four real-world clinical datasets. ConSurv achieves such gains in the discriminative performance while providing the best or comparable calibration performance. Notably, as evidenced by the metrics IBS, DDC, and D-CAL, our method yields exceptional calibration performance. Contrarily, ranking loss-based models such as DeepHit and DRSA, which are designed primarily to enhance discriminative power, usually exhibit poor calibration. Even without being specifically designed through the loss function for calibration, our model outperforms benchmarks such as DCS and X-CAL, which incorporate loss functions used to enhance calibration power. This indicates that our approach inherently balances both discriminative and calibration power effectively.

**Ablation Study.** We conduct an ablation study to better understand the contribution of different loss functions within our framework employing the following variants of ConSurv: (i) $\mathcal{L}_{NLL}$ only, (ii) $\mathcal{L}_{NLL}$ & $\mathcal{L}_{NCE}$, (iii) $\mathcal{L}_{NLL}$ & $\mathcal{L}_{Rank}$ , and (iv) ConSurv (i.e., $\mathcal{L}_{NLL}$ & $\mathcal{L}_{SNCE}$). Table 1 highlights the performance trade-offs between discrimination and calibration power when different loss functions are utilized in survival models. Using only NLL loss typically leads to lower discriminative performance while maintaining reasonably good calibration. Augmenting a ranking loss to the NLL significantly enhances discriminative performance but decreases calibration. In contrast, our proposed SNCE loss significantly enhances the model's performance compared to (i) using NLL only and (iii) using NLL with ranking loss. This suggests that the SNCE loss not only boosts discriminative power but also preserves or even improves the model's calibration. Additionally, when compared to (ii) using NLL with InfoNCE, which pushes all negative pairs away equally, we confirm that ConSurv, which assigns weights to negative samples based on their time difference until the event, ensures better performance.

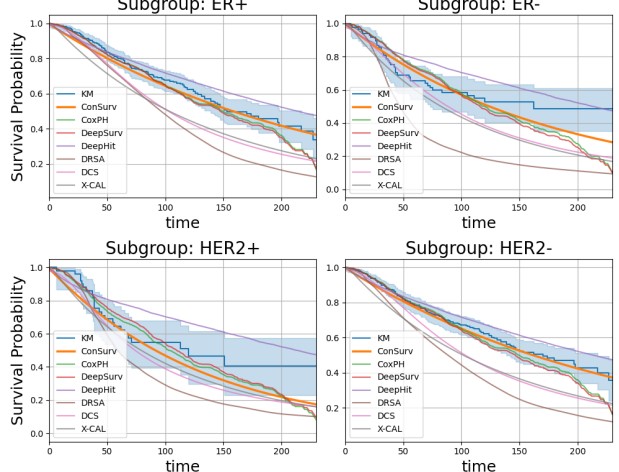

Table 2: Wasserstein distances from KM curves across various patient subgroups for the METABRIC dataset.

| Subgroup | ER | | HER2 | |
|---|---|---|---|---|
| | + | - | + | - |
| CoxPH | 0.030 | 0.108 | 0.063 | **0.089** |
| DeepSurv | 0.033 | 0.115 | 0.066 | 0.101 |
| DeepHit | 0.063 | 0.082 | 0.146 | 0.156 |
| DRSA | 0.181 | 0.293 | 0.233 | 0.328 |
| DCS | 0.130 | 0.146 | 0.087 | 0.178 |
| X-CAL | 0.136 | 0.165 | 0.105 | 0.180 |
| ConSurv | **0.024** | **0.077** | **0.044** | **0.089** |

Figure 4: Comparison of the survival curves across various patient subgroups for the METABRIC dataset.

## 5.3 Qualitative Analysis Results

### 5.3.1 Effect of Contrastive Learning

To further demonstrate how survival outcome-aware contrastive learning promotes discrimination among samples in the latent space, we compare two-dimensional t-SNE visualizations of latent representations trained with $\mathcal{L}_{NLL}$ only, $\mathcal{L}_{SNCE}$ only and ConSurv (i.e., $\mathcal{L}_{NLL}$ & $\mathcal{L}_{SNCE}$) for the METABRIC dataset in Figure 2. Here, we only display uncensored samples with event times for clarity. Figure 2 shows that when trained with $\mathcal{L}_{NLL}$ alone, latent representations are mixed regardless of their corresponding event times. However, incorporating $\mathcal{L}_{SNCE}$ into the loss function significantly improves the alignment of representations with event time information, enabling the predictor to better discriminate the risks associated with each sample. Notably, even when trained solely with $\mathcal{L}_{SNCE}$, latent representations exhibit a clear alignment with event times. This finding strongly suggests that our contrastive learning approach effectively encourages discrimination by capturing and reflecting the underlying event time information. (See Appendix C.2 for more datasets.)

### 5.3.2 Calibration Plot

Figure 3 shows calibration plots comparing the calibration of ConSurv with the deep learning-based survival models. Here, we evaluate calibration by matching predicted cumulative densities to observed event frequencies at the quantiles of the predicted cumulative density. In these plots, the x=y line represents the ideal state where predicted probabilities perfectly match the observed outcomes. The evaluated survival models show different trends depending on the loss function used for training. The ranking-based models (i.e., DeepHit and DRSA), which directly manipulate the model outputs, experimentally demonstrated the lowest calibration power. Contrarily, the calibration-based models (i.e., X-CAL and DCS), designed to improve calibration power, show relatively better calibration but do not match the calibration performance of DeepSurv, likely due to the assumption of proportional hazards. Compared to these survival models trained based on three different types of loss functions, our proposed method demonstrates calibration performance that is similar or even superior.

### 5.3.3 Subgroup Analysis

We further validate the calibration performance of the survival models by comparing their survival plots with the Kaplan-Meier (KM) curve, a non-parametric estimate of the survival function at a population level. For this comparison, we examine two binary hormone receptor statuses in the METABRIC dataset: estrogen receptor (ER) and human epidermal growth factor receptor 2 (HER2), which are crucial for determining hormone therapies and can result in significantly different prognoses for breast cancer patients. In Figure 4, we average the survival functions of different survival models based on the subgroups of patients with the corresponding feature values (e.g., ER+ vs ER-). The

results confirm that our method aligns well with the KM curve compared to other deep learning-based survival models, especially those with ranking loss.

To quantitatively assess calibration performance across different subgroups, we compare the survival predictions of each model with KM curves for each subgroup using the Wasserstein distance. This metric is well-suited for comparing survival curves as it considers both the overall shape and the discrepancies in predicted survival probabilities at each time point. The results in Table 2 demonstrate that our proposed model achieves well-calibrated predictions within each subgroup. (See Appendix C.4.2 for more subgroups.)

## 6 Conclusion

In this paper, we propose ConSurv, a novel contrastive learning approach for deep survival analysis. Our method leverages weighted sampling to incorporate the intuitive assumption that patients with similar event times likely share similar clinical statuses. Unlike most existing deep survival models, ConSurv does not directly manipulate hazard or survival predictions during contrastive learning. This approach allows our method to maintain well-calibrated predictions based on the negative log-likelihood loss. Experiments on multiple real-world datasets demonstrate the superiority of our method over state-of-the-art deep survival models, particularly in achieving superior calibration performance compared to ranking loss-based models.

## Acknowledgments

This work was supported by the National Research Foundation of Korea (NRF) grant funded by the Korea government (MSIT) (No. RS-2024-00358602) and the Institute of Information & Communications Technology Planning & Evaluation (IITP) grant funded by the Korea government (MSIT) (No. RS-2019-II190079, Artificial Intelligence Graduate School Program (Korea University).

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

# A Notations

The notations used in this work are listed in Table 3.

Table 3: Table of Notations.

| Notation | Description |
|---|---|
| $A_{i,j}$ | The indicator for acceptable pairs in the ranking loss |
| $\alpha$ | The margin to ensure a minimum time difference |
| $\beta$ | The balancing coefficient between the loss function |
| $\delta_i$ | The Event indicator for patient $i$ |
| $\mathcal{D}$ | The survival dataset |
| $f_\phi$ | The hazard network function |
| $f_\theta$ | The encoder function |
| $g_\psi$ | The projection head function |
| $\mathcal{H}$ | The latent representation space |
| $\lambda$ | The conditional hazard function |
| $\mathcal{L}_{NLL}$ | The negative log-likelihood loss |
| $\mathcal{L}_{Rank}$ | The ranking loss |
| $\mathcal{L}_{SNCE}$ | The survival outcome-based contrastive learning loss |
| $M$ | The number of samples in a batch |
| $R$ | The risk function |
| $S$ | The survival function |
| $\sigma$ | The temperature coefficient for the weight function |
| $\nu$ | The temperature coefficient for the SNCE loss |
| $\mathcal{T}$ | The set of all possible discrete survival times |
| $\tau_i$ | The observed survival time for patient $i$ |
| $T_{\max}$ | The maximum time horizon |
| $\tilde{\mathbf{x}}_i$ | The corrupted version of the original sample $\mathbf{x}_i$ |
| $w$ | The weight function |
| $Z$ | The normalization constant |
| $\mathbf{x}_i$ | The input feature vector for patient $i$ |
| $\mathcal{X}$ | The input feature space |

# B Augmentation Method

In our experiments with tabular datasets, we compared augmentation methods, focusing on traditional noise injection (NI) and the state-of-the-art method, SCARF [26]. For noise injection, we introduced random Gaussian noise into the latent space, determining the optimal corruption rate through a hyperparameter search consistent with SCARF. SCARF consistently showed performance that was similar to or better than NI. Consequently, we adopted SCARF as our primary augmentation method for our experiments.

Table 4: Ablation study on the different tabular augmentation methods.

| | METABRIC | | | | NWTCO | | | |
|---|---|---|---|---|---|---|---|---|
| **METHOD** | CI↑ | IBS↓ | DDC↓ | D-CAL↑ | CI↑ | IBS↓ | DDC↓ | D-CAL↑ |
| NI | 0.650 | 0.096 | **0.052** | 0.003 | 0.704 | **0.059** | 0.250 | 0.137 |
| SCARF | **0.688** | **0.094** | 0.097 | 0.089 | **0.729** | 0.078 | **0.222** | 0.360 |
| | GBSG | | | | FLCHAIN | | | |
| **METHOD** | CI↑ | IBS↓ | DDC↓ | D-CAL↑ | CI↑ | IBS↓ | DDC↓ | D-CAL↑ |
| NI | 0.681 | 0.157 | **0.070** | 0.000 | 0.788 | 0.088 | **0.247** | 0.096 |
| SCARF | **0.696** | **0.138** | 0.180 | 0.004 | **0.810** | **0.084** | 0.291 | 0.204 |
| | SUPPORT | | | | SEER | | | |
| **METHOD** | CI↑ | IBS↓ | DDC↓ | D-CAL↑ | CI↑ | IBS↓ | DDC↓ | D-CAL↑ |
| NI | 0.612 | 0.204 | **0.149** | 0.000 | 0.842 | 0.014 | 0.975 | 0.999 |
| SCARF | **0.625** | **0.195** | 0.155 | 0.000 | **0.854** | **0.009** | **0.960** | 0.999 |

# C  Additional Results

Table 5: Comparison of time-dependent C-index.

| Methods | METABRIC | | | NWTCO | | |
|---|---|---|---|---|---|---|
| | 25% | 50% | 75% | 25% | 50% | 75% |
| CoxPH | $0.653_{\pm0.024}$ | $0.645_{\pm0.018}$ | $0.637_{\pm0.018}$ | $0.718_{\pm0.026}$ | $0.711_{\pm0.025}$ | $0.706_{\pm0.026}$ |
| DeepSurv | $0.643_{\pm0.034}$ | $0.620_{\pm0.024}$ | $0.607_{\pm0.029}$ | $0.640_{\pm0.081}$ | $0.636_{\pm0.079}$ | $0.634_{\pm0.078}$ |
| DeepHit | $0.605_{\pm0.045}$ | $0.659_{\pm0.044}$ | $0.603_{\pm0.032}$ | $0.717_{\pm0.023}$ | $\mathbf{0.723_{\pm0.023}}$ | $0.704_{\pm0.024}$ |
| DRSA | $0.639_{\pm0.030}$ | $0.605_{\pm0.020}$ | $0.592_{\pm0.022}$ | $0.713_{\pm0.020}$ | $0.706_{\pm0.019}$ | $0.704_{\pm0.019}$ |
| DCS | $0.646_{\pm0.030}$ | $0.615_{\pm0.023}$ | $0.606_{\pm0.024}$ | $0.675_{\pm0.026}$ | $0.657_{\pm0.020}$ | $0.643_{\pm0.029}$ |
| X-CAL | $0.661_{\pm0.025}$ | $0.632_{\pm0.018}$ | $0.619_{\pm0.019}$ | $0.687_{\pm0.026}$ | $0.650_{\pm0.029}$ | $0.628_{\pm0.034}$ |
| **ConSurv** | $\mathbf{0.686_{\pm0.028}}$ | $\mathbf{0.660_{\pm0.021}}$ | $\mathbf{0.645_{\pm0.023}}$ | $\mathbf{0.718_{\pm0.025}}$ | $0.709_{\pm0.024}$ | $\mathbf{0.705_{\pm0.025}}$ |

| Methods | GBSG | | | FLCHAIN | | |
|---|---|---|---|---|---|---|
| | 25% | 50% | 75% | 25% | 50% | 75% |
| CoxPH | $0.714_{\pm0.027}$ | $0.678_{\pm0.019}$ | $0.664_{\pm0.018}$ | $0.796_{\pm0.019}$ | $\mathbf{0.793_{\pm0.015}}$ | $0.790_{\pm0.014}$ |
| DeepSurv | $0.700_{\pm0.056}$ | $0.668_{\pm0.047}$ | $0.654_{\pm0.043}$ | $0.787_{\pm0.028}$ | $0.783_{\pm0.027}$ | $0.780_{\pm0.028}$ |
| DeepHit | $0.630_{\pm0.045}$ | $0.626_{\pm0.030}$ | $0.609_{\pm0.028}$ | $0.796_{\pm0.020}$ | $\mathbf{0.793_{\pm0.014}}$ | $0.785_{\pm0.014}$ |
| DRSA | $0.725_{\pm0.687}$ | $0.687_{\pm0.015}$ | $0.671_{\pm0.016}$ | $0.776_{\pm0.023}$ | $0.776_{\pm0.016}$ | $0.772_{\pm0.013}$ |
| DCS | $0.720_{\pm0.025}$ | $0.687_{\pm0.015}$ | $0.674_{\pm0.017}$ | $0.788_{\pm0.024}$ | $0.783_{\pm0.019}$ | $0.779_{\pm0.018}$ |
| X-CAL | $\mathbf{0.727_{\pm0.025}}$ | $0.680_{\pm0.016}$ | $0.676_{\pm0.017}$ | $0.793_{\pm0.021}$ | $0.788_{\pm0.015}$ | $0.784_{\pm0.016}$ |
| **ConSurv** | $0.723_{\pm0.030}$ | $\mathbf{0.689_{\pm0.019}}$ | $\mathbf{0.676_{\pm0.020}}$ | $\mathbf{0.796_{\pm0.022}}$ | $0.791_{\pm0.017}$ | $\mathbf{0.786_{\pm0.015}}$ |

| Methods | SUPPORT | | | SEER | | |
|---|---|---|---|---|---|---|
| | 25% | 50% | 75% | 25% | 50% | 75% |
| CoxPH | $0.603_{\pm0.006}$ | $0.607_{\pm0.006}$ | $0.609_{\pm0.006}$ | $0.865_{\pm0.018}$ | $0.890_{\pm0.023}$ | $0.855_{\pm0.022}$ |
| DeepSurv | $0.603_{\pm0.009}$ | $0.605_{\pm0.009}$ | $0.608_{\pm0.010}$ | $0.838_{\pm0.007}$ | $0.835_{\pm0.021}$ | $0.836_{\pm0.002}$ |
| DeepHit | $0.516_{\pm0.007}$ | $0.489_{\pm0.008}$ | $0.521_{\pm0.007}$ | $0.847_{\pm0.016}$ | $0.841_{\pm0.020}$ | $0.827_{\pm0.020}$ |
| DRSA | $0.586_{\pm0.009}$ | $0.583_{\pm0.007}$ | $0.580_{\pm0.009}$ | $0.840_{\pm0.073}$ | $0.839_{\pm0.077}$ | $0.820_{\pm0.083}$ |
| DCS | $0.604_{\pm0.008}$ | $0.602_{\pm0.009}$ | $0.602_{\pm0.009}$ | $0.864_{\pm0.019}$ | $0.863_{\pm0.023}$ | $0.858_{\pm0.023}$ |
| X-CAL | $0.610_{\pm0.006}$ | $0.608_{\pm0.006}$ | $0.608_{\pm0.007}$ | $0.840_{\pm0.023}$ | $0.839_{\pm0.019}$ | $0.833_{\pm0.018}$ |
| **ConSurv** | $\mathbf{0.616_{\pm0.006}}$ | $\mathbf{0.617_{\pm0.006}}$ | $\mathbf{0.618_{\pm0.006}}$ | $\mathbf{0.870_{\pm0.017}}$ | $\mathbf{0.866_{\pm0.020}}$ | $\mathbf{0.862_{\pm0.020}}$ |

## C.1  Performance on Time-dependent C-index and Brier score

While the CI and IBS provide valuable overall assessments of a given survival model (over the entire time horizon), these metrics cannot fully capture variations in model performance across different time points. To address this, we included time-dependent performance evaluations, namely the time-dependent C-index [5] and the time-dependent Brier score[30] at three different time points (i.e., 25%, 50%, and 75% percentiles of time-to-events as in [32, 33, 34]. It is worth highlighting that utilizing these time-dependent metrics may reveal subtle differences that might be obscured when using CI and IBS alone. As shown in Table 5 and 6, we demonstrate the superiority of our method in discriminative power while preserving calibration. To evaluate the performance of survival models at various time points, we select the 25%, 50%, and 75%-percentile of each dataset.

## C.2  Effect of Contrastive Learning

In Figure 5, When using $\mathcal{L}_{NLL}$ alone, representations tend to cluster tightly in a narrow space regardless of time. However, when applying $\mathcal{L}_{SNCE}$ (contrastive learning with a time weight) to $\mathcal{L}_{NLL}$, representations of features with similar times are more likely to be positioned close to each other.

## C.3  Sensitive Analysis

**Sensitive Analysis on Coefficient** $\alpha$.  $\alpha$ is the margin to ensure a minimum time difference between samples in acceptable pairs. All the weights for the entire sample are precomputed. For setting the margin, we calculate the percentile for only the weights corresponding to **Case 2** mentioned in section

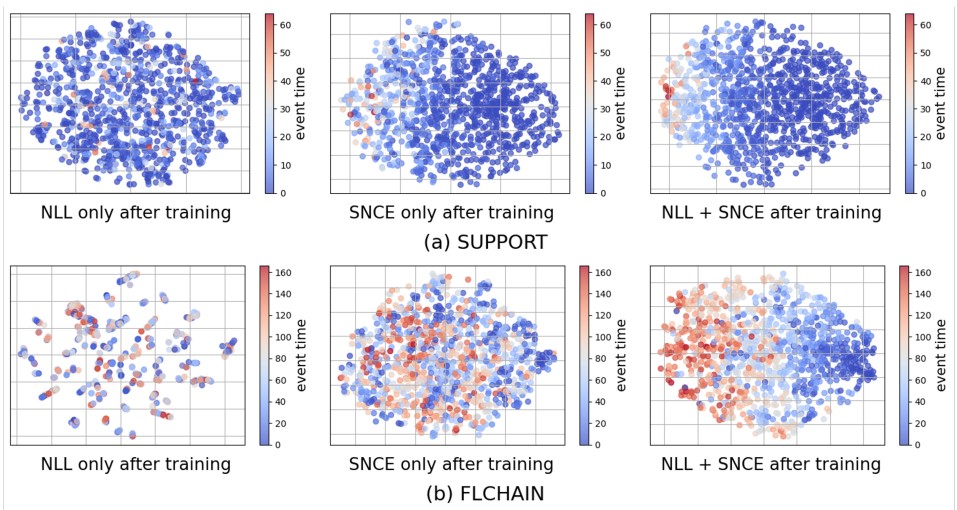

Figure 5: t-SNE visualization for latent representations learned with $\mathcal{L}_{NLL}$ only, $\mathcal{L}_{SNCE}$ only, and ConSurv for the FLCHAIN and SUPPORT datasets, colored by event times (for uncensored samples).

Table 6: Comparison of time-dependent Brier score.

| Methods | METABRIC | | | NWTCO | | |
|---|---|---|---|---|---|---|
| | 25% | 50% | 75% | 25% | 50% | 75% |
| CoxPH | $0.167_{\pm 0.009}$ | $\mathbf{0.215_{\pm 0.014}}$ | $\mathbf{0.236_{\pm 0.038}}$ | $\mathbf{0.111_{\pm 0.009}}$ | $0.114_{\pm 0.008}$ | $0.114_{\pm 0.008}$ |
| DeepSurv | $0.173_{\pm 0.010}$ | $0.228_{\pm 0.016}$ | $0.249_{\pm 0.048}$ | $0.121_{\pm 0.011}$ | $0.124_{\pm 0.011}$ | $0.125_{\pm 0.012}$ |
| DeepHit | $0.181_{\pm 0.009}$ | $0.256_{\pm 0.010}$ | $0.263_{\pm 0.026}$ | $0.151_{\pm 0.009}$ | $0.143_{\pm 0.008}$ | $0.154_{\pm 0.009}$ |
| DRSA | $0.264_{\pm 0.041}$ | $0.343_{\pm 0.038}$ | $0.337_{\pm 0.079}$ | $0.281_{\pm 0.030}$ | $0.294_{\pm 0.039}$ | $0.310_{\pm 0.051}$ |
| DCS | $0.190_{\pm 0.020}$ | $0.265_{\pm 0.038}$ | $0.293_{\pm 0.083}$ | $0.119_{\pm 0.010}$ | $0.107_{\pm 0.011}$ | $0.122_{\pm 0.024}$ |
| X-CAL | $0.180_{\pm 0.012}$ | $0.234_{\pm 0.021}$ | $0.254_{\pm 0.057}$ | $0.113_{\pm 0.009}$ | $\mathbf{0.106_{\pm 0.013}}$ | $0.137_{\pm 0.036}$ |
| **ConSurv** | $\mathbf{0.166_{\pm 0.009}}$ | $0.221_{\pm 0.015}$ | $0.255_{\pm 0.045}$ | $0.111_{\pm 0.009}$ | $0.113_{\pm 0.008}$ | $\mathbf{0.113_{\pm 0.009}}$ |

| Methods | GBSG | | | FLCHAIN | | |
|---|---|---|---|---|---|---|
| | 25% | 50% | 75% | 25% | 50% | 75% |
| CoxPH | $\mathbf{0.145_{\pm 0.010}}$ | $0.211_{\pm 0.007}$ | $0.219_{\pm 0.010}$ | $\mathbf{0.074_{\pm 0.006}}$ | $\mathbf{0.112_{\pm 0.006}}$ | $\mathbf{0.133_{\pm 0.007}}$ |
| DeepSurv | $0.146_{\pm 0.011}$ | $0.212_{\pm 0.009}$ | $0.221_{\pm 0.013}$ | $0.076_{\pm 0.007}$ | $0.114_{\pm 0.009}$ | $0.137_{\pm 0.011}$ |
| DeepHit | $0.168_{\pm 0.012}$ | $0.242_{\pm 0.008}$ | $0.250_{\pm 0.006}$ | $0.096_{\pm 0.005}$ | $0.181_{\pm 0.004}$ | $0.244_{\pm 0.004}$ |
| DRSA | $0.210_{\pm 0.016}$ | $0.322_{\pm 0.022}$ | $0.349_{\pm 0.020}$ | $0.141_{\pm 0.013}$ | $0.251_{\pm 0.027}$ | $0.323_{\pm 0.031}$ |
| DCS | $0.153_{\pm 0.010}$ | $0.210_{\pm 0.006}$ | $0.218_{\pm 0.010}$ | $0.079_{\pm 0.008}$ | $0.119_{\pm 0.008}$ | $0.147_{\pm 0.014}$ |
| X-CAL | $0.150_{\pm 0.011}$ | $\mathbf{0.207_{\pm 0.007}}$ | $\mathbf{0.217_{\pm 0.012}}$ | $0.080_{\pm 0.005}$ | $0.117_{\pm 0.007}$ | $0.144_{\pm 0.012}$ |
| **ConSurv** | $0.146_{\pm 0.009}$ | $0.209_{\pm 0.008}$ | $\mathbf{0.217_{\pm 0.010}}$ | $0.077_{\pm 0.006}$ | $0.115_{\pm 0.007}$ | $0.136_{\pm 0.008}$ |

| Methods | SUPPORT | | | SEER | | |
|---|---|---|---|---|---|---|
| | 25% | 50% | 75% | 25% | 50% | 75% |
| CoxPH | $\mathbf{0.219_{\pm 0.003}}$ | $0.197_{\pm 0.008}$ | $0.180_{\pm 0.009}$ | $\mathbf{0.005_{\pm 0.001}}$ | $\mathbf{0.009_{\pm 0.001}}$ | $\mathbf{0.014_{\pm 0.001}}$ |
| DeepSurv | $0.220_{\pm 0.004}$ | $0.198_{\pm 0.009}$ | $0.180_{\pm 0.010}$ | $\mathbf{0.005_{\pm 0.001}}$ | $0.010_{\pm 0.001}$ | $0.016_{\pm 0.001}$ |
| DeepHit | $0.340_{\pm 0.005}$ | $0.282_{\pm 0.003}$ | $0.240_{\pm 0.005}$ | $0.016_{\pm 0.000}$ | $0.020_{\pm 0.001}$ | $0.026_{\pm 0.001}$ |
| DRSA | $0.285_{\pm 0.013}$ | $0.269_{\pm 0.022}$ | $0.249_{\pm 0.021}$ | $0.011_{\pm 0.006}$ | $0.021_{\pm 0.016}$ | $0.034_{\pm 0.026}$ |
| DCS | $0.222_{\pm 0.006}$ | $0.209_{\pm 0.015}$ | $0.202_{\pm 0.017}$ | $\mathbf{0.005_{\pm 0.001}}$ | $0.010_{\pm 0.001}$ | $0.015_{\pm 0.002}$ |
| X-CAL | $0.220_{\pm 0.005}$ | $0.207_{\pm 0.016}$ | $0.201_{\pm 0.018}$ | $0.006_{\pm 0.001}$ | $0.011_{\pm 0.002}$ | $0.019_{\pm 0.003}$ |
| **ConSurv** | $\mathbf{0.219_{\pm 0.005}}$ | $\mathbf{0.195_{\pm 0.010}}$ | $\mathbf{0.177_{\pm 0.010}}$ | $\mathbf{0.005_{\pm 0.001}}$ | $\mathbf{0.009_{\pm 0.001}}$ | $0.015_{\pm 0.001}$ |

4.2. By defining this minimum time difference, the margin aids in reducing the ambiguity in survival outcomes, leading to a more precise comparison of patient risks. Specifically, this enhancement is vital for improving the model's discrimination capabilities, as it removes potential false negative samples with little difference in their survival outcomes. In Table 7, to determine the sensitivity of the ConSurv with respect to the value of $\alpha$, we report the performance results on four datasets by varying

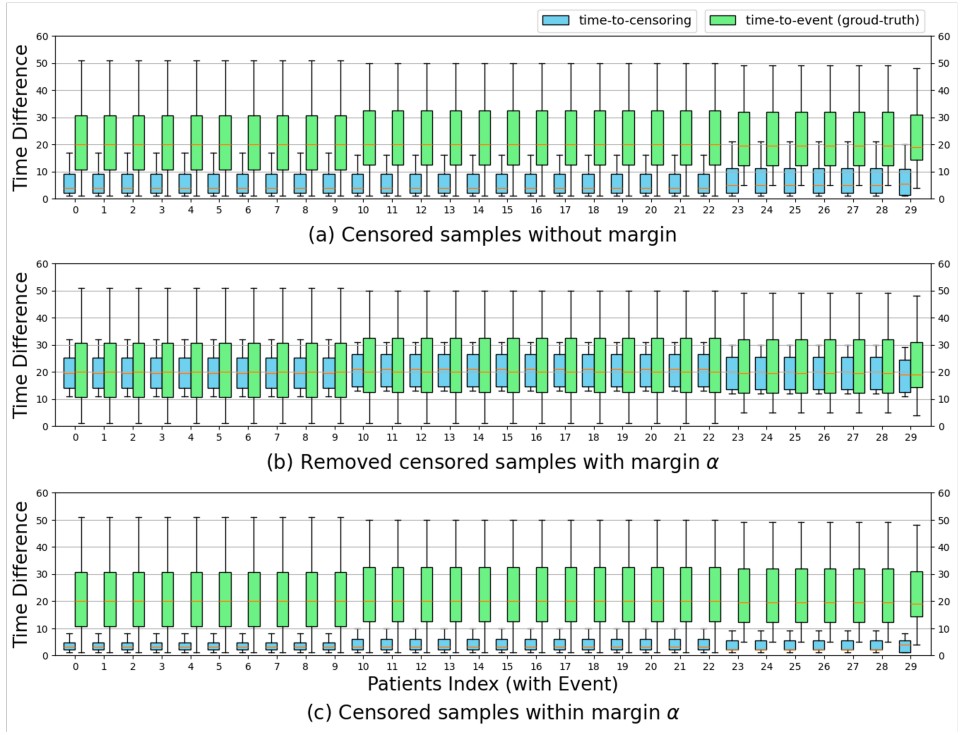

Figure 6: Comparison of time differences based on censoring times and those based on the (unobserved) ground-truth event times. Time differences are computed for each sample with an event with respect to the censoring time and event time of its corresponding comparable censored samples. (Samples are listed in ascending order of their event times.)

the value of $\alpha$ according to the following percentiles of time differences: $\{5.0, 7.0, 10.0, 12.0, 14.0\}$. Additionally, we compare the performance metrics with no margin (i.e., $\alpha = 0$).

**Effect of Margin $\alpha$.**  In Figure 6, we compare time differences based on censoring times and the (unobserved) ground-truth event times. For each event sample, we compute the time differences between its event time and: 1) the (observed) censoring time of its corresponding comparable censored sample, and 2) the (unobserved) ground-truth event time of that same censored sample.

To show this effect, we have conducted an experiment using additional synthetic data by adapting the time-to-event generation process in DeepHit [1] for both event times $T_i$ and censoring times $C_i$:

$$T_i \sim \exp\left((10x_{i_1})^2 + 5x_{i_3}\right), \quad C_i \sim \exp\left((10x_{i_2})^2 + 5x_{i_4}\right) \quad (11)$$

Here, we randomly generated 1,000 random samples, where a sample is censored when $T_i > C_i$.

Without a margin, censored samples with censoring times close to event times may be incorrectly treated as having similar event times, leading to very low weight assignments. To avoid this, we introduce a margin to exclude such samples when computing our contrastive loss. Overall, the margin in $|\tau_i - \tau_j| \geq \alpha$ helps prevent failures in accurately distinguishing risk between comparable censored samples.

**Sensitive Analysis On Coefficient $\beta$.**  $\beta$ is the balancing coefficient in our overall loss function where increased $\beta$ makes the impact of $\mathcal{L}_{SNCE}$ more dominant. Table 8 shows that when $\beta$ (using the optimal $\alpha$ for each dataset) is relatively small (here, $\beta = 0.01$), contrastive loss struggles to effectively learn similarities between survival outcomes, leading to poor discrimination performance. However, increasing the $\beta$ ratio consistently improves performance. This suggests that our proposed contrastive learning approach does not prevent the survival model from learning effectively guided by the NLL loss.

Table 7: Impact of margin ($\alpha$) on performance metrics.

| METHOD | METABRIC | | | | NWTCO | | | |
|---|---|---|---|---|---|---|---|---|
| | CI↑ | IBS↓ | DDC↓ | D-CAL↑ | CI↑ | IBS↓ | DDC↓ | D-CAL↑ |
| $\alpha = 0$ | 0.695 | 0.088 | 0.100 | 0.256 | 0.765 | 0.065 | 0.481 | 0.001 |
| $\alpha = 5$ | 0.687 | 0.089 | 0.098 | 0.243 | 0.751 | 0.068 | 0.480 | 0.012 |
| $\alpha = 7$ | 0.691 | 0.088 | 0.083 | 0.134 | **0.773** | **0.059** | **0.463** | 0.503 |
| $\alpha = 10$ | **0.707** | **0.084** | **0.093** | 0.528 | 0.756 | 0.069 | 0.573 | 0.061 |
| $\alpha = 12$ | 0.692 | 0.088 | 0.083 | 0.671 | 0.751 | 0.067 | 0.497 | 0.703 |
| $\alpha = 14$ | 0.692 | 0.086 | 0.064 | 0.230 | 0.753 | 0.069 | 0.506 | 0.159 |

| METHOD | GBSG | | | | FLCHAIN | | | |
|---|---|---|---|---|---|---|---|---|
| | CI↑ | IBS↓ | DDC↓ | D-CAL↑ | CI↑ | IBS↓ | DDC↓ | D-CAL↑ |
| $\alpha = 0$ | 0.691 | 0.153 | 0.201 | 0.000 | 0.779 | 0.100 | 0.306 | 0.731 |
| $\alpha = 5$ | 0.693 | 0.153 | 0.199 | 0.000 | 0.771 | 0.100 | **0.254** | 0.929 |
| $\alpha = 7$ | **0.695** | **0.151** | **0.182** | 0.000 | **0.783** | **0.090** | 0.280 | 0.693 |
| $\alpha = 10$ | 0.694 | 0.154 | 0.224 | 0.000 | 0.779 | 0.100 | 0.270 | 0.963 |
| $\alpha = 12$ | 0.673 | 0.157 | 0.241 | 0.000 | 0.778 | 0.095 | 0.258 | 0.919 |
| $\alpha = 14$ | 0.670 | 0.159 | 0.236 | 0.000 | 0.777 | 0.098 | 0.310 | 0.929 |

| METHOD | SUPPORT | | | | SEER | | | |
|---|---|---|---|---|---|---|---|---|
| | CI↑ | IBS↓ | DDC↓ | D-CAL↑ | CI↑ | IBS↓ | DDC↓ | D-CAL↑ |
| $\alpha = 0$ | 0.592 | 0.205 | 0.150 | 0.000 | 0.883 | 0.008 | 0.962 | 0.999 |
| $\alpha = 5$ | 0.605 | 0.198 | **0.142** | 0.000 | 0.886 | 0.007 | 0.948 | 0.999 |
| $\alpha = 7$ | 0.604 | 0.202 | 0.145 | 0.000 | 0.874 | 0.012 | 0.984 | 0.991 |
| $\alpha = 10$ | **0.610** | **0.196** | 0.144 | 0.000 | **0.889** | **0.004** | **0.933** | 0.999 |
| $\alpha = 12$ | 0.608 | 0.197 | 0.147 | 0.000 | 0.879 | 0.008 | 0.954 | 0.999 |
| $\alpha = 14$ | 0.598 | 0.203 | 0.149 | 0.000 | 0.877 | 0.007 | 0.952 | 0.999 |

Table 8: Impact of margin ($\beta$) on performance metrics.

| METHOD | METABRIC | | | | NWTCO | | | |
|---|---|---|---|---|---|---|---|---|
| | CI↑ | IBS↓ | DDC↓ | D-CAL↑ | CI↑ | IBS↓ | DDC↓ | D-CAL↑ |
| $\beta = 0.01$ | 0.673 | 0.089 | 0.148 | 0.254 | 0.747 | **0.069** | 0.493 | 0.043 |
| $\beta = 0.1$ | 0.682 | 0.124 | 0.109 | 0.320 | 0.759 | 0.091 | 0.470 | 0.095 |
| $\beta = 1.0$ | **0.706** | **0.086** | 0.093 | 0.524 | **0.773** | 0.078 | 0.222 | 0.428 |
| $\beta = 10.0$ | 0.652 | 0.166 | 0.096 | 0.405 | 0.759 | 0.093 | 0.344 | 0.397 |
| $\beta = 100.0$ | 0.743 | 0.071 | **0.463** | 0.352 | 0.743 | 0.071 | **0.463** | 0.352 |

| METHOD | GBSG | | | | FLCHAIN | | | |
|---|---|---|---|---|---|---|---|---|
| | CI↑ | IBS↓ | DDC↓ | D-CAL↑ | CI↑ | IBS↓ | DDC↓ | D-CAL↑ |
| $\beta = 0.01$ | 0.655 | 0.149 | 0.150 | 0.015 | 0.746 | **0.063** | 0.493 | 0.353 |
| $\beta = 0.1$ | 0.677 | 0.171 | 0.156 | 0.012 | 0.775 | 0.108 | 0.364 | 0.543 |
| $\beta = 1.0$ | **0.703** | **0.138** | 0.180 | 0.075 | **0.816** | 0.096 | **0.338** | 0.465 |
| $\beta = 10.0$ | 0.686 | 0.169 | **0.139** | 0.002 | 0.777 | 0.107 | 0.354 | 0.537 |
| $\beta = 100.0$ | 0.694 | 0.170 | 0.363 | 0.057 | 0.743 | 0.071 | 0.551 | 0.057 |

| METHOD | SUPPORT | | | | SEER | | | |
|---|---|---|---|---|---|---|---|---|
| | CI↑ | IBS↓ | DDC↓ | D-CAL↑ | CI↑ | IBS↓ | DDC↓ | D-CAL↑ |
| $\beta = 0.01$ | 0.594 | 0.180 | 0.134 | 0.000 | 0.831 | 0.016 | 0.983 | 0.999 |
| $\beta = 0.1$ | 0.603 | 0.185 | 0.139 | 0.000 | 0.842 | 0.020 | 0.982 | 0.999 |
| $\beta = 1.0$ | **0.610** | 0.183 | **0.132** | 0.000 | **0.854** | **0.010** | **0.975** | 0.999 |
| $\beta = 10.0$ | 0.605 | 0.194 | 0.145 | 0.000 | 0.832 | 0.016 | 0.984 | 0.999 |
| $\beta = 100.0$ | 0.601 | **0.179** | 0.152 | 0.000 | 0.821 | 0.021 | 0.988 | 0.999 |

## C.4 Calibration

### C.4.1 Calibration Plot

Figure 7 shows calibration plots of ConSurv in comparison with the benchmarks on all the evaluated datasets. Here, similar to our observation in Section 5.3.2, ConSurv consistently exhibits calibration performance that is similar to or better than the benchmarks, effectively balancing discriminative and calibration power.

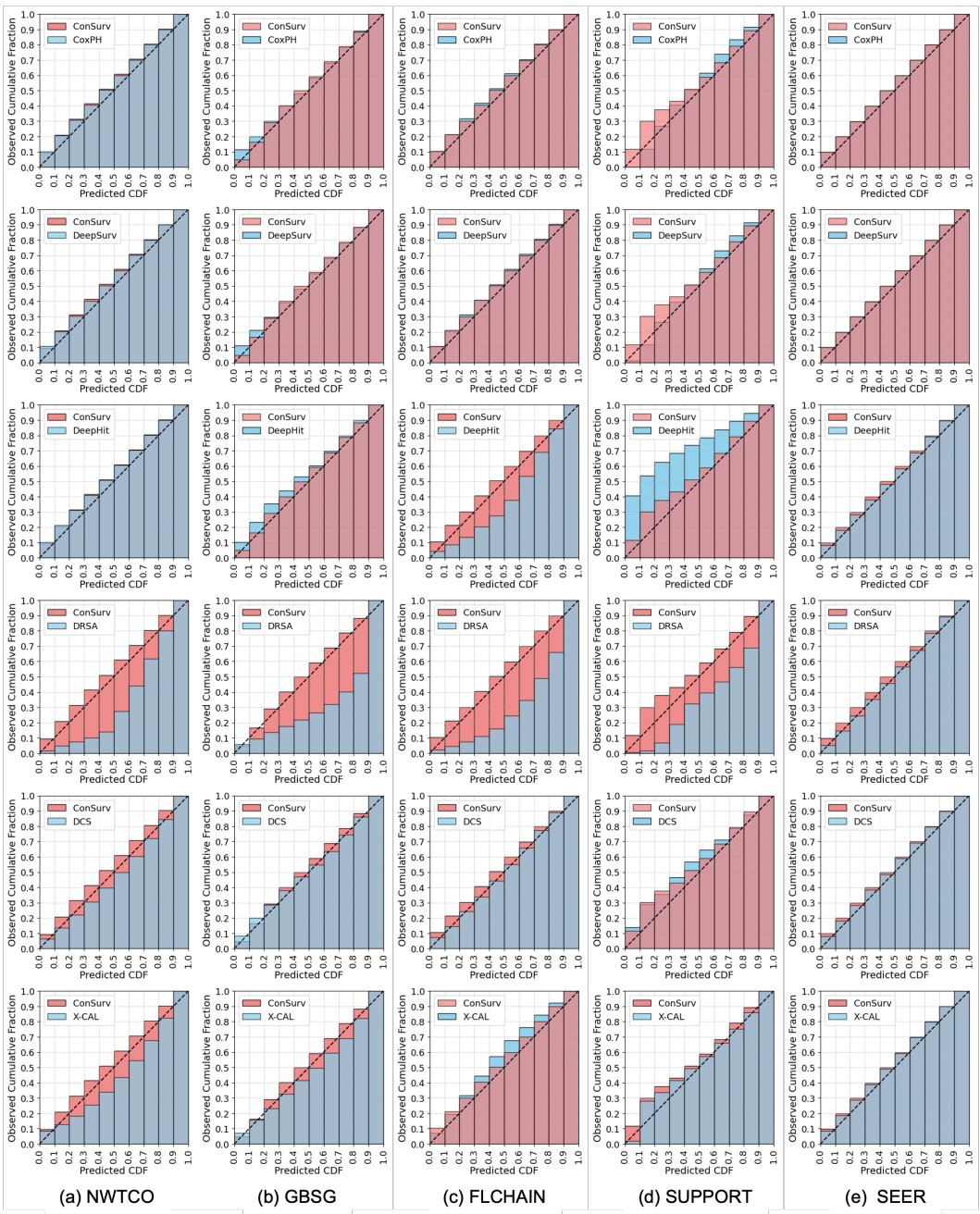

Figure 7: Calibration plots for ConSurv in comparison with benchmarks.

### C.4.2 Subgroup Analysis

In this section, we show that our model learns well about the differences between subgroups on real-world datasets, with good calibration performance by comparing the averaged survival predictions and the observed KM estimation within each subgroup. In Figure 8, we subgroup patients based on cellularity, which is a categorical variable representing the degree of density of cells in the tumor. Similarly, in Figure 9 (a), we partition patients based on the size of the tumor (with + indicating 70% or more and - indicating 30% or less of the total size), which have a significant impact on the event. In Figure 9 (b), we examine binary hormone receptor statuses progesterone receptor (PR).

Our results show that our method closely aligns with the KM curve, outperforming other deep learning-based survival models, especially those with ranking loss. We assess calibration performance using the Wasserstein distance to compare survival predictions with KM curves across different subgroups. In Table 9 indicate that our model provides well-calibrated predictions within each subgroup.

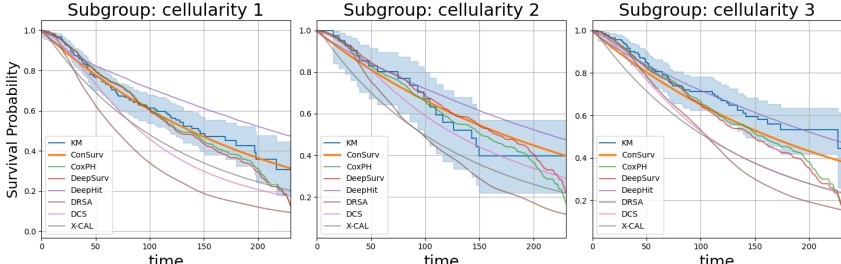

Figure 8: Comparison of the survival curves across various patient subgroups in the METABRIC dataset.

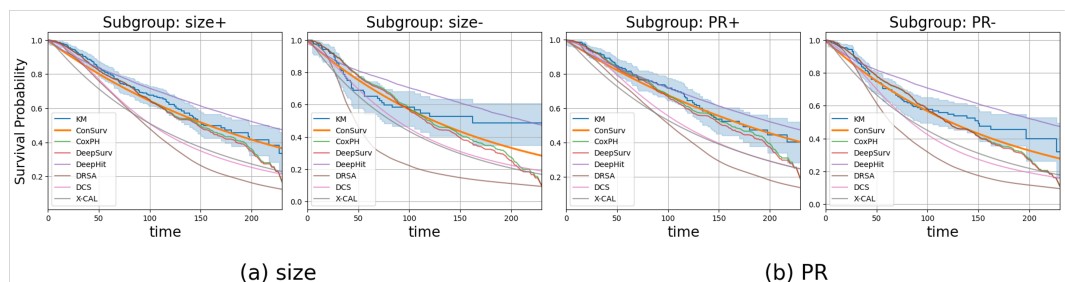

Figure 9: Comparison of the survival curves across various patient subgroups in the METABRIC dataset.

Table 9: Subgroup-level Wasserstein distances on the METABRIC dataset.

| Subgroup | cellularity | | | PR | | size | |
|---|---|---|---|---|---|---|---|
| | 1 | 2 | 3 | + | - | + | - |
| CoxPH | 0.039 | **0.044** | 0.067 | 0.049 | 0.067 | 0.076 | 0.044 |
| DeepSurv | 0.044 | 0.048 | 0.080 | 0.054 | 0.074 | 0.105 | 0.089 |
| DeepHit | 0.101 | 0.074 | **0.020** | **0.043** | 0.102 | 0.233 | 0.033 |
| DRSA | 0.215 | 0.168 | 0.203 | 0.205 | 0.276 | 0.118 | 0.234 |
| DCS | 0.125 | 0.060 | 0.168 | 0.154 | 0.126 | 0.091 | 0.124 |
| X-CAL | 0.113 | 0.143 | 0.184 | 0.159 | 0.148 | **0.060** | 0.176 |
| ConSurv | **0.018** | **0.044** | 0.063 | **0.043** | **0.042** | **0.060** | **0.025** |

## C.5 Evaluating Performance Across Extended Seeds

For enhanced stability verification, we conducted additional experiments with 25 random seeds in Table 10.

Table 10: Discrimination and calibration of survival models: mean and standard deviation values for CI, IBS, and DDC, along with the number of successful D-calibration tests.

| | METABRIC | | | | NWTCO | | | |
|---|---|---|---|---|---|---|---|---|
| **METHOD** | CI ↑ | IBS ↓ | DDC ↓ | D-CAL | CI ↑ | IBS ↓ | DDC ↓ | D-CAL |
| CoxPH | $0.645_{\pm0.019}$ | $\mathbf{0.175_{\pm0.028}}$ | $0.111_{\pm0.024}$ | **25** | $0.716_{\pm0.025}$ | $0.108_{\pm0.008}$ | $0.515_{\pm0.022}$ | **25** |
| DeepSurv | $0.625_{\pm0.025}$ | $0.183_{\pm0.029}$ | $0.103_{\pm0.026}$ | **25** | $0.640_{\pm0.080}$ | $0.117_{\pm0.011}$ | $0.792_{\pm0.011}$ | 24 |
| DeepHit | $0.604_{\pm0.019}$ | $0.204_{\pm0.018}$ | $0.292_{\pm0.017}$ | 0 | $0.717_{\pm0.028}$ | $0.143_{\pm0.024}$ | $0.657_{\pm0.024}$ | 12 |
| DRSA | $0.604_{\pm0.032}$ | $0.249_{\pm0.038}$ | $0.178_{\pm0.060}$ | 0 | $0.709_{\pm0.019}$ | $0.281_{\pm0.041}$ | $0.218_{\pm0.065}$ | 0 |
| DCS | $0.612_{\pm0.029}$ | $0.206_{\pm0.043}$ | $\mathbf{0.054_{\pm0.039}}$ | 2 | $0.642_{\pm0.036}$ | $0.119_{\pm0.018}$ | $0.209_{\pm0.043}$ | 19 |
| X-CAL | $0.632_{\pm0.027}$ | $0.182_{\pm0.023}$ | $0.065_{\pm0.037}$ | 2 | $0.622_{\pm0.037}$ | $0.128_{\pm0.025}$ | $\mathbf{0.191_{\pm0.079}}$ | 12 |
| $\mathcal{L}_{NLL}$ | $0.642_{\pm0.022}$ | $0.197_{\pm0.030}$ | $0.077_{\pm0.020}$ | 13 | $0.707_{\pm0.024}$ | $0.109_{\pm0.008}$ | $0.556_{\pm0.041}$ | 23 |
| $\mathcal{L}_{NLL}$ & $\mathcal{L}_{NCE}$ | $0.659_{\pm0.020}$ | $0.193_{\pm0.029}$ | $0.080_{\pm0.022}$ | 21 | $0.715_{\pm0.024}$ | $0.108_{\pm0.009}$ | $0.563_{\pm0.054}$ | 22 |
| $\mathcal{L}_{NLL}$ & $\mathcal{L}_{Rank}$ | $0.652_{\pm0.022}$ | $0.247_{\pm0.030}$ | $0.177_{\pm0.020}$ | 0 | $0.717_{\pm0.027}$ | $0.137_{\pm0.008}$ | $0.653_{\pm0.050}$ | 0 |
| **ConSurv** | $\mathbf{0.665_{\pm0.023}}$ | $0.186_{\pm0.021}$ | $0.110_{\pm0.024}$ | 23 | $\mathbf{0.718_{\pm0.025}}$ | $\mathbf{0.107_{\pm0.008}}$ | $0.554_{\pm0.045}$ | 24 |

| | GBSG | | | | FLCHAIN | | | |
|---|---|---|---|---|---|---|---|---|
| **METHOD** | CI ↑ | IBS ↓ | DDC ↓ | D-CAL | CI ↑ | IBS ↓ | DDC ↓ | D-CAL |
| CoxPH | $0.662_{\pm0.179}$ | $0.181_{\pm0.007}$ | $0.183_{\pm0.037}$ | **25** | $0.790_{\pm0.012}$ | $\mathbf{0.103_{\pm0.005}}$ | $0.287_{\pm0.013}$ | **25** |
| DeepSurv | $0.653_{\pm0.042}$ | $0.182_{\pm0.009}$ | $0.153_{\pm0.066}$ | 24 | $0.779_{\pm0.028}$ | $0.106_{\pm0.006}$ | $0.287_{\pm0.021}$ | **25** |
| DeepHit | $0.633_{\pm0.032}$ | $0.205_{\pm0.006}$ | $0.342_{\pm0.023}$ | 3 | $0.795_{\pm0.023}$ | $0.170_{\pm0.005}$ | $0.492_{\pm0.018}$ | 0 |
| DRSA | $0.668_{\pm0.016}$ | $0.278_{\pm0.018}$ | $0.402_{\pm0.055}$ | 0 | $0.772_{\pm0.014}$ | $0.230_{\pm0.024}$ | $0.369_{\pm0.034}$ | 0 |
| DCS | $\mathbf{0.677_{\pm0.017}}$ | $0.181_{\pm0.008}$ | $\mathbf{0.124_{\pm0.025}}$ | 10 | $0.779_{\pm0.016}$ | $0.113_{\pm0.009}$ | $0.363_{\pm0.136}$ | 5 |
| X-CAL | $0.675_{\pm0.017}$ | $0.181_{\pm0.010}$ | $0.166_{\pm0.020}$ | 8 | $0.783_{\pm0.015}$ | $0.112_{\pm0.008}$ | $0.402_{\pm0.105}$ | 3 |
| $\mathcal{L}_{NLL}$ | $0.668_{\pm0.019}$ | $0.179_{\pm0.006}$ | $0.154_{\pm0.029}$ | 0 | $0.783_{\pm0.014}$ | $0.106_{\pm0.004}$ | $0.264_{\pm0.028}$ | 19 |
| $\mathcal{L}_{NLL}$ & $\mathcal{L}_{NCE}$ | $0.669_{\pm0.020}$ | $0.179_{\pm0.007}$ | $0.155_{\pm0.026}$ | 0 | $0.786_{\pm0.013}$ | $0.106_{\pm0.004}$ | $0.283_{\pm0.042}$ | 22 |
| $\mathcal{L}_{NLL}$ & $\mathcal{L}_{Rank}$ | $0.687_{\pm0.019}$ | $0.280_{\pm0.007}$ | $0.263_{\pm0.025}$ | 0 | $0.785_{\pm0.014}$ | $0.146_{\pm0.005}$ | $0.303_{\pm0.024}$ | 0 |
| **ConSurv** | $\mathbf{0.677_{\pm0.020}}$ | $\mathbf{0.179_{\pm0.007}}$ | $0.160_{\pm0.026}$ | 18 | $\mathbf{0.796_{\pm0.014}}$ | $0.105_{\pm0.004}$ | $\mathbf{0.280_{\pm0.036}}$ | 23 |

| | SUPPORT | | | | SEER | | | |
|---|---|---|---|---|---|---|---|---|
| **METHOD** | CI ↑ | IBS ↓ | DDC ↓ | D-CAL | CI ↑ | IBS ↓ | DDC ↓ | D-CAL |
| CoxPH | $0.604_{\pm0.006}$ | $0.191_{\pm0.005}$ | $0.262_{\pm0.013}$ | 0 | $0.858_{\pm0.018}$ | $\mathbf{0.009_{\pm0.005}}$ | $0.966_{\pm0.003}$ | **25** |
| DeepSurv | $0.603_{\pm0.090}$ | $0.192_{\pm0.007}$ | $0.245_{\pm0.036}$ | 0 | $0.814_{\pm0.020}$ | $0.010_{\pm0.000}$ | $1.000_{\pm0.000}$ | **25** |
| DeepHit | $0.503_{\pm0.009}$ | $0.272_{\pm0.003}$ | $0.337_{\pm0.006}$ | 0 | $0.840_{\pm0.033}$ | $0.020_{\pm0.001}$ | $0.836_{\pm0.003}$ | 0 |
| DRSA | $0.570_{\pm0.009}$ | $0.259_{\pm0.015}$ | $0.486_{\pm0.084}$ | 0 | $0.834_{\pm0.078}$ | $0.021_{\pm0.015}$ | $\mathbf{0.671_{\pm0.135}}$ | 0 |
| DCS | $0.598_{\pm0.008}$ | $0.207_{\pm0.012}$ | $0.175_{\pm0.032}$ | 0 | $0.860_{\pm0.020}$ | $0.010_{\pm0.001}$ | $0.911_{\pm0.044}$ | 21 |
| X-CAL | $0.603_{\pm0.007}$ | $0.204_{\pm0.012}$ | $0.181_{\pm0.025}$ | 0 | $0.837_{\pm0.040}$ | $0.015_{\pm0.006}$ | $0.900_{\pm0.049}$ | 18 |
| $\mathcal{L}_{NLL}$ | $0.606_{\pm0.006}$ | $0.193_{\pm0.008}$ | $0.123_{\pm0.019}$ | 0 | $0.854_{\pm0.016}$ | $0.009_{\pm0.001}$ | $0.867_{\pm0.009}$ | 25 |
| $\mathcal{L}_{NLL}$ & $\mathcal{L}_{NCE}$ | $0.608_{\pm0.007}$ | $0.192_{\pm0.007}$ | $0.127_{\pm0.021}$ | 0 | $0.859_{\pm0.017}$ | $0.012_{\pm0.003}$ | $0.964_{\pm0.006}$ | 25 |
| $\mathcal{L}_{NLL}$ & $\mathcal{L}_{Rank}$ | $0.617_{\pm0.007}$ | $0.173_{\pm0.008}$ | $0.231_{\pm0.024}$ | 0 | $0.862_{\pm0.017}$ | $0.139_{\pm0.001}$ | $1.000_{\pm0.000}$ | 0 |
| **ConSurv** | $\mathbf{0.616_{\pm0.005}}$ | $\mathbf{0.190_{\pm0.006}}$ | $0.148_{\pm0.023}$ | 0 | $\mathbf{0.864_{\pm0.016}}$ | $0.009_{\pm0.001}$ | $0.863_{\pm0.006}$ | **25** |

# D  Dataset Description and Experimental Setup

## D.1  Real-World Datasets

Table 11: Descriptive statistics of the real-world datasets

| Dataset | No. Uncensored | No. Censored | No. Features (real, binary, categorical) |
|---|---|---|---|
| METABRIC | 888 (55.2%) | 1093 (44.8%) | 21(6,0,15) |
| NWTCO | 571 (14.2%) | 3457 (85.8%) | 6 (1,4,1) |
| GBSG | 1267 (56.8%) | 965 (43.2%) | 7 (4,2,1) |
| FLCHAIN | 4562 (69.9%) | 1962 (30.0%) | 8(4,2,2) |
| SUPPORT | 6036 (68.1%) | 2837 (31.9%) | 14 (8,3,3) |
| SEER | 604(1.11%) | 53940(98.9%) | 12(4,5,3) |

Table 11 shows baseline statistics of the real-world datasets. ***METABRIC*** [35] comprises clinical features and gene expression profiles (overall 21 features) used to determine the risk of breast cancer

subgroups for 1,981 patients. Among the total number of patients, 1,093 (55.2%) are censored, and 888 (44.8%) are uncensored. **NWTCO** [36] is a study related to Wilms' tumor, containing 4,023 patients with 6 clinical and histology features. Among the total number of patients, 3,457(85.8%) are censored and 571(14.2%) are uncensored. **GBSG** [37] investigates the effects of hormone treatment on recurrence-free survival time, with the event of interest being breast cancer recurrence. The dataset consists of 21 clinical and tumor-related features for 2,232 patients. Among them, 965 (43.2%) are censored and 1,267 (56.8%) are uncensored. **FLCHAIN** [38] contains subjects from a study of the relationship between serum-free light chain and mortality, where each subject is described by 8 features. Among 6,524 subjects, 1,962 (30.1%) are censored and 4,562 (69.9%) are uncensored. **SUPPORT** [39] focuses on the survival time of seriously ill hospitalized adults. The dataset consists of 14 clinical features for 8,873 patients, having 2,837 (31.9%) censored and 6,036 (68.1%) uncensored. **SEER** [40] provides data on prostate cancer patients, containing 12 clinical and cancer-related features for 54,544 patients. Among them, 604 (1.11%) are uncensored and 53,940 (98.9%) are censored. We split the data into train, test, and validation sets with a ratio of 0.64:0.20:0.16, and then apply min-max normalization to the input features.

## D.2 Performance Metrics

**Integrated Brier Score (IBS)** is metric to evaluate the mean squared error between the predicted survival curve with step function of the observed event. The IBS is the integration of the Brier score across all time points.

**Distributional Divergence for Calibration (DDC)** is a novel metric introduced in [2] to assess the calibration quality of survival models. This metric computes the divergence between the empirical distribution of estimated survival probabilities at the observed event times and a uniform distribution. Specifically, DDC utilizes the Kullback-Leibler (KL) Divergence between these two distributions.

- It first bins the estimated survival probabilities $\hat{S}(t_i|x_i)$ at the observed event times into ten equal-width intervals across the [0,1] range.
- It then calculates the KL divergence $D_{KL}(P \parallel Q)$ between this binned distribution $P$ and a theoretical uniform distribution $Q$.
- DDC is scaled so that its values range between 0 and 1, with lower values indicating better calibration. A perfectly calibrated model, where the estimated probabilities perfectly match a uniform distribution, would yield a DDC value of 0.

DDC serves as a more nuanced measure compared to traditional metrics like the Brier score or C-index because it directly evaluates how closely the distribution of the model's probabilistic predictions matches the expected uniform distribution of true probabilities. This direct assessment helps in determining whether the model's predictions are systematically biased or well-aligned with actual outcomes.

**D-Calibration (D-CAL)** assesses how well predicted survival probabilities align with observed outcomes based on a goodness-of-fit test. D-CAL bins the predicted survival probabilities at the true event times into ten equal-width intervals from 0 to 1, and performs a chi-squared test to ascertain the uniformity of the distribution. As suggested in [31], we reports $p$-values from the D-CAL test and the number of datasets with $p$-values greater than 0.05.

## D.3 About Benchmarks

**Benchmarks.** We compare ConSurv with six survival models including one traditional statistical method and four state-of-the-art deep learning methods: **CoxPH** [27] is a commonly used statistical method under the proportional hazard assumption[4], **DeepSurv** [28] is an extension of the Cox model which utilizes a DNN to predict individual hazard rates[5], **DeepHit** [1] is a DNN model which directly models the joint distribution of the event times[6], **DRSA** [10] is an RNN-based model which predicts the likelihood of the true event occurrence and estimates the survival rate over time[7], **DCS** [2] is an RNN-based model that estimates calibrated individualized survival curves while maximizing the

---

[4]Python package `scikit-survival`
[5]`https://github.com/havakv/pycox`
[6]`https://github.com/chl8856/DeepHit`
[7]`https://github.com/rk2900/DRSA`

discriminative power[8], and ***X-CAL*** [15] is a survival model designed to focus on the calibration performance by directly utilizing a differentiable version of D-CAL as a part of the objectives. [9] Especially with ***DCS*** and ***X-CAL***, we made every effort to faithfully reproduce the code with hyperparameter optimization, as the original code had several errors and we were unable to use it for our experiments.

# E   Implementation

We discuss implementation of ConSurv below. The source code for ConSurv is available in https://github.com/dongzza97/ConSurv. Throughout the experiments, training ConSurv and its variants takes approximately 60 minutes to 2 hours on a single GPU machine.[10]

## E.1   Hyperparameter Specification

We perform a random search for hyperparameter optimization – including the batch size, hidden dimension, depth, learning rates, corruption rates, $\sigma$, $\alpha$, and $\nu$ – on the validation set and choose the settings with the best performance for ConSurv on each dataset. Table 12 describes the model specifications for the evaluated datasets. The candidates for random search can be given as {32, 64, 128, 256}, {16, 32, 64}, and {3, 4, 5} for batch size, hidden dim, and depth respectively. We carefully choose the $\sigma$, $\alpha$ and $\nu$ for $\mathcal{L}_{SNCE}$ over {0.25, 0.5, 0.75}, {5.0, 7.0, 10.0, 12.0, 14.0} and {0.03, 0.05, 0.07}. Corruption rates are selected from a range of 0.1 to 0.9. To train multiple components in our network simultaneously, we need to find different learning rates for each network $f_\theta, g_\psi$, and $f_\phi$, where the learning rate is chosen from a set {1e-3, 1e-4}.

Table 12: Hyperparameter specifications

| Dataset | Hidden Dim | Batch Size | Depth | Learning Rate | Corruption Rate | $\sigma$ | $\alpha$ | $\nu$ |
|---|---|---|---|---|---|---|---|---|
| METABRIC | 32 | 16 | 4 | 1e-4, 1e-4 | 0.5 | 0.75 | 10.0 | 0.07 |
| NWTCO | 8 | 32 | 5 | 1e-4, 1e-3 | 0.7 | 0.75 | 7.0 | 0.05 |
| GBSG | 8 | 32 | 5 | 1e-4, 1e-3 | 0.7 | 0.75 | 7.0 | 0.05 |
| FLCHAIN | 16 | 128 | 3 | 1e-3, 1e-3 | 0.8 | 0.25 | 7.0 | 0.03 |
| SUPPORT | 32 | 64 | 4 | 1e-3, 1e-3 | 0.7 | 0.25 | 10.0 | 0.07 |
| SEER | 32 | 256 | 5 | 1e-4, 1e-4 | 0.4 | 0.75 | 0.0 | 0.03 |

# F   Broader Impact

In this work, we highlight the direct and indirect influence of our approach through experiments on real-world clinical datasets. The use of these datasets was in accordance with the guidance of the respective data providers and domain experts. We acknowledge the potential for both positive and negative impacts associated with survival analysis techniques. While these methods can provide valuable insights for personalized treatment and risk stratification, we emphasize the importance of careful interpretation and validation by domain experts before applying these predictions in clinical practice. Misuse or misunderstanding of survival analysis could lead to unintended consequences for individuals or groups, such as inequitable access to care or discriminatory practices. Therefore, we advocate for transparency, accountability, and ongoing dialogue among researchers, clinicians, and the public to ensure the responsible and ethical use of survival analysis in healthcare.

# G   Limitation and Future Work

Survival analysis datasets often include features where a clear event time is not available, indicating a significant number of censored patients. This makes it challenging to assign weights based on time differences and to compare them against each other. Therefore, utilizing such data for learning

---

[8]https://github.com/MLD3/Calibrated-Survival-Analysis

[9]https://github.com/rajesh-lab/X-CAL

[10]The specification of the machine is CPU: INTEL XEON Gold 6240R, GPU: NVIDIA RTX A6000

purposes presents difficulties. Overcoming these limitations requires further research. Potential avenues include modifying models to account for such uncertainties or developing alternative learning approaches. Additionally, there is a need for new evaluation metrics or model developments that consider the characteristics of such data.

Furthermore, the tabular datasets we utilized often lack diverse augmentation techniques. Future research could explore augmentation methods suitable for survival datasets to enhance contrastive learning performance. For example, techniques considering temporal variations or reflecting uncertainties in events could be investigated. Such research endeavors are expected to enhance understanding in the fields of survival analysis and contrastive learning, contributing to improved model performance.

## H   Pseudo-code for ConSurv

---

**Algorithm 1** Pseudo-code for the ConSurv

---

**Input:** Dataset $\mathcal{D} = \{(\mathbf{x}_i, \tau_i, \delta_i)\}_{i=1}^{N}$, batch size $M$, margin $\alpha$, coefficient $\beta$
       encoder $\theta$, projection head $\psi$, hazard network $\phi$, learning rate $\eta_1, \eta_2$.
**Output:** ConSurv parameters $(\theta, \psi, \phi)$

Initialize parameters $(\theta, \psi, \phi)$
**repeat**
   Sample a mini-batch of $\{(\mathbf{x}_i, \tau_i, \delta_i)\}_{i=1}^{M} \sim \mathcal{D}$
   **for** $i = 1, ..., M$ **do**
      Draw augmentation function: $s \sim \mathcal{S}$
      Augmentation pair: $\mathbf{x}_i, \mathbf{x}_i^{+} \leftarrow \mathbf{x}_i, s(\mathbf{x}_i)$
      Latent representation: $\mathbf{h}_i, \mathbf{h}_i^{+} \leftarrow f_\theta(\mathbf{x}_i), f_\theta(\mathbf{x}_i^{+})$
      Map to the embedding space: $\mathbf{z}_i, \mathbf{z}_i^{+} \leftarrow g_\psi(\mathbf{h}_i), g_\psi(\mathbf{h}_i^{+})$
   **end for**
   **for all** $i = 1 \cdots, 2M$ and $j = 1 \cdots, 2M$ **do**
      Cosine similarity in the embedding space: $s(\mathbf{x}_i, \mathbf{x}_j) = \mathbf{z}_i^{\top} \mathbf{z}_j / (\|\mathbf{z}_i\|\|\mathbf{z}_j\|)$
      Weight function for negative distribution: $w(\tau_j^{-}; \tau_i) = 1 - e^{\frac{-|\tau_i - \tau_j^{-}|}{\sigma}}$
      Redefined weight function considering the right-censoring: $\tilde{w}(\tau_j^{-}; \tau_i) = I_{i,j} \cdot w(\tau_j^{-}; \tau_i)$
   **end for**
   Update the encoder and projection head parameter $\theta, \psi$:

$$(\theta, \psi) \leftarrow (\theta, \psi) - \eta_1 \nabla_{(\theta, \psi)} \left( \frac{1}{2M} \sum_{i=1}^{2M} \beta \cdot \left[ -\log \frac{e^{s(\mathbf{x}_i, \mathbf{x}_i^{+})}}{\frac{1}{Z} \sum_{j=1}^{2M} \mathbb{I}_{[i \neq j]} \tilde{w}(\mathbf{x}_j^{-}; \mathbf{x}_i) \cdot e^{s(\mathbf{x}_i, \mathbf{x}_j^{-})}} \right] \right)$$

   **for** $i = 1, ..., M$ **do**
      Compute latent representations: $\mathbf{h}_i \leftarrow f_\theta(\mathbf{x}_i)$
      Compute estimates for survival function: $\hat{S}_i(\tau_i|\mathbf{x}_i) \leftarrow \prod_{t \leq \tau_i}(1 - f_\phi(\mathbf{h}_i, t))$
      Compute estimates for probability mass
      function: $\hat{p}_i(\tau_i|\mathbf{x}_i) \leftarrow f_\phi(\mathbf{h}_i, \tau_i) \prod_{t < \tau_i}(1 - f_\phi(\mathbf{h}_i, t))$
   **end for**

   Update the encoder and hazard network parameters $(\theta, \phi)$:

$$(\theta, \phi) \leftarrow (\theta, \phi) - \eta_2 \nabla_{(\theta, \phi)} \left( -\frac{1}{M} \sum_{i=1}^{M} \delta_i \log \hat{p}(\tau_i|\mathbf{x}_i) + (1 - \delta_i) \log \hat{S}(\tau_i|\mathbf{x}_i) \right)$$

**until** convergence

---

