# OpenReview forum: "Toward a Well-Calibrated Discrimination via Survival Outcome-Aware Contrastive Learning"
_NeurIPS.cc/2024/Conference — NeurIPS 2024 poster_

### Official Review · Reviewer_VjbS · 2024-07-05

**Soundness:** 2
**Presentation:** 3
**Contribution:** 3
**Rating:** 6
**Confidence:** 4

**Summary:**

This work concerns deep learning for survival analysis. Deep learning models for survival analysis are typically desired to have good discriminative performance, where a model can differentiate between patients of different risk profile, and calibration performance, where the time-to-event is accurately predicted by the model. Methods which seek to improve discrimination performance have often led to poorer model calibration. Consequently, this paper proposes an auxiliary contrastive loss to improve the discriminative performance of deep survival models, while still achieving high calibration performance. The novelty in this method is adapting the SOTA contrastive learning method for tabular data, SCARF, by weighting the negative pairs according to the difference in time-to-event. Hence, samples with greater difference in time-to-event are considered more important negative pairs, enforcing an inductive bias to differentiate the latent representations of samples with different time-to-event.  After providing a detailed problem formulation, the contrastive loss and generation of positive and negative pairs is described. The results of several experiments are provided which compare the proposed method with current state-of-the-art methods when applied to a number of real world datasets, and demonstrate the proposed model performance under ablation of each term in the loss function. From these experiments it is concluded that the auxiliary contrastive loss achieves superior discriminative and calibration performance.

**Strengths:**

The strength of this work lies in its originality, quality and clarity. The paper proposes an interesting adaptation to SCARF, a contrastive method for tabular data, by weighting the negative pairs such that samples with greater difference in time-to-event are considered more important negative pairs. Recent work on contrastive learning and deep survival methods are cited and appropriately discussed.

Technically, the paper is sound, with a good mathematical foundation for the importance sampling in the weighted contrastive loss term, and a thorough set of experiments which establishes performance against important baselines and provides good support for modelling choices via ablation studies. Furthermore, the clarity of the paper is for the most part good: the description of the background, methodology, and experiments are clear and detailed.

**Weaknesses:**

While the technical parts of this paper are clearly explained, the organisation and presentation of the paper could be improved. In particular, the introduction uses quite a lot of terminology specific to survival analysis that is unexplained or explained in detail only later in the paper. As a specific example, censoring is first mentioned on line 67, but is not clearly explained. Explaining this terminology, which may not be familiar to all of NeurIPS wide readership, at first use would improve the clarity of the paper. Presentation and clarity of the paper would also be improved if all captions were self-contained. In some cases, for example in Table 1, it is difficult to interpret the results as presented, and this could have been avoided if the captions were more detailed. Specifically, it is not clear what the standard deviations represent in this table i.e. variation across model initialisations or data splits.

This work should be praised for providing a list of experiments that should provide the evidence needed to make the claim that the proposed model provides superior discriminative performance while maintaining calibration performance. However, the clarity and quality of these experiments is sometimes below expectations, and this ultimately reduces the overall significance of this work. Table 1 is used to report that the proposed model is the best performing model. However, it seems that the model is within 1 standard deviation of the second best performing model, and hence the statistical significance of this result is unclear. As previously mentioned, while mean and standard deviation of performance metrics are reported, it is not entirely clear what distribution they refer to – whether it be different data splits, or model initialisations via random seed. This makes it difficult to interpret the significance of these results, and also makes it more difficult to reproduce.

The paper, therefore, addresses an important problem, proposes a novel and well thought out method to address this issue, but the significance is undermined by a lack of clarity and ambiguity surrounding the statistical significance of reported results.

**Questions:**

The questions below focus on the details of the experiments that were unclear. Clarity on these questions may change the overall opinion of the paper.

Table 1: In the caption you refer to the standard deviation - are the reported values in this table the mean?
Table 1: Does the standard deviation refer to variation over different data splits, model initialisations or something else?
Table 1: What do bold and underlined values represent?
Table 1: How do you determine that the superior performance of ConSurv is statistically significant?

Line 528: Can you confirm whether min-max normalisation was performed before or after data was split into train, test and validation sets?

Line 236: What is the marginal corruption process? By what augmentation were positive pairs generated?

Line 197: What does the temperature coefficient scale? What impact does this have on model training?

Line 104: Does ref. [18] show that model training is destabilised by the contrastive loss? If not, what supports this statement? Could this model have been included in your benchmarks?


NOTE: Due to the discussion in the author-reviewer discussion period, I feel that many of these points were clarified. In particular new results have better demonstrated the statistical significance of the performance of ConSurv. I raise my score from a 4 to a 6 to reflect this.

**Limitations:**

The authors have addressed the potential negative societal impact of their work, stating that model predictions from such work should be under scrutiny by domain experts before being applied in a clinical setting. They stress that incorrect use could lead to inequitable access or discriminatory practices. While this is welcome, it is not specific to this work, but is a concern of all works proposing machine learning for clinical practice. A potential impact more specific to this work might consider why the inductive bias introduced by the auxiliary contrastive loss would lead to unintended consequences for AI-supported decisions on treatment and intervention. For example, the unsupervised nature of the contrastive learning procedure could align patients based on features which may act as a proxy for protected traits and lead to inequitable model performance amongst different demographics.

Additionally, a discussion of limitations is provided but is limited. The authors correctly identify that a significant number of patients are censored in real world datasets, and this undermines the importance sampling based on difference in time-to-event, as weights are assigned according to a function in equation (5). It would improve the discussion of the limitations if it was pointed out that seeking alternatives or amendments to (5) would be a specific avenue to improve upon this work.

The limitation section could benefit from greater specificity. The authors propose “modifying models to account for such uncertainties or developing alternative learning methods”. The uncertainties they refer to are unclear, and a preferred statement might be: future avenues of work would be to develop a model which retains the benefit of contrastive learning with respect to discriminative performance, whilst also being agnostic to the number of censored samples within the training set.

They then state “there is a need for new evaluation metrics or model developments that consider the characteristics of such data”. The clarity of this proposal might be clearer if stated with more specificity i.e. "to develop a model that retains high discriminatory and calibration performance for datasets with arbitrary proportions of censored samples, new metrics must be established that can compare the same model applied to different proportions of censored samples."

---

> ### Author Rebuttal · Authors · 2024-08-07
>
> # [ Response to Reviewer VjbS ]
> We thank the reviewer for the valuable suggestions on our work. We have addressed the reviewer’s comments in our response below.
> ## (Details of Table 1)
> - Mean and Standard Deviation: The values in Table 1 represent the mean performance metrics, with standard deviations provided in Appendix C.1, Table 5. These standard deviations show the variation across 10 random data splits for each model, ensuring robustness and reliability. We have updated the manuscript to clearly refer to the appendix for details on standard deviation.
> - Meaning of Bold and Underlined values: Bold values indicate the models with the best performance for each dataset. Underlined values indicate where the D-CAL measure is above 0.05, meaning that calibration is guaranteed.
> - Statistical Significance of ConSurv’s Performance: To assess the statistical significance of the performance comparison, we conducted Welch’s t-test. In Table E, we present the findings where the improvement was statistically significant (i.e., p-value < 0.05). We have provided a detailed explanation and statistical analysis to substantiate our claim in the updated manuscript.
>
> *Table E. p-value for Performance Metrics.*
> |Model|Metric|METABRIC|NWTCO|GBSG|FLCHAIN|SUPPORT|SEER|
> |-|-|-|-|-|-|-|-|
> |CoxPH|CI|0.004|0.264|0.047|0.325|0.001|0.047|
> ||IBS|0.204|0.022|0.612|0.033|0.315|0.000|
> |DeepSurv|CI|0.031|0.013|0.009|0.115|0.003|0.000|
> ||IBS|0.302|0.034|0.487|0.046|0.229|0.000|
> |DeepHit|CI|0.020|0.385|0.010|0.301|0.436|0.404|
> ||IBS|0.110|0.000|0.000|0.300|0.000|0.000|
> |DRSA|CI|0.000|0.185|0.385|0.000|0.000|0.050|
> ||IBS|0.000|0.000|0.000|0.000|0.000|0.000|
> |DCS|CI|0.000|0.000|0.346|0.040|0.030|0.191|
> ||IBS|0.000|0.240|0.312|0.040|0.000|0.000|
> |X-CAL|CI|0.025|0.000|0.491|0.128|0.050|0.050|
> ||IBS|0.012|0.025|0.212|0.044|0.060|0.000|
> ## (Details of Generating Positive/Negative Samples)
> We construct the positive and negative samples based on the contrastive learning framework for tabular data [R8]. Specifically, we generate augmented versions $\tilde{x}^{(i)}$ of each data point $x^{(i)}$ by randomly selecting a subset of features (up to a pre-specified corruption rate) and replacing the value by sampling the corresponding feature's marginal distribution. The augmented version of the reference sample is used as the positive pair, and the augmented versions of the remaining samples in the mini-batch are used as the negative pairs. This corruption process is applied regardless of the feature types. We have included the details of generating the positive and negative samples in the updated manuscript.
> ## (Additional Experiments: New Baselines)
> In Table F, we have included a new baseline SupWcon which is a variant of our method where we replace $\mathcal{L}_{SNCE}$ with a supervised contrastive loss that directly weights negative samples based on the actual time differences of the comparable pairs, without considering their distributional properties of these weights [R11]. This can make weighting of negative samples sensitive to the scale of the time differences, potentially reducing the robustness of the contrastive learning loss. Notably, while weighted contrastive learning based on the difference in time-to-events show slight performance gains, the improvement is not significant when compared to ours. This suggests the importance of applying good inductive bias when contrasting samples, potentially because it encourages the model to focus on discriminating the survival prediction based on a proper distribution defined based on the differences in time-to-events.
>
> *Table F. Discrimination and calibration of survival models for ConSurv w/ $\mathcal{L}_{SupWcon}$.*
> |Data|CI|IBS|DDC|D-Cal|
> |-|-|-|-|-|
> |METABRIC|0.637±0.025|0.198±0.032|0.106±0.029|0.098±0.109|
> |NWTCO|0.721±0.042|0.106±0.012|0.558±0.055|0.571±0.483|
> |GBSG|0.675±0.027|0.186±0.018|0.179±0.025|0.000±0.000|
> |FLCHAIN|0.781±0.021|0.105±0.006|0.314±0.053|0.332±0.394|
> ## (Temperature coefficient scale)
> We thank the reviewer for insightful feedback regarding the use of $\sigma$ in our time-to-event contrastive learning framework. Here, we explain how $\sigma$ values affect model performance.
> When $\sigma$ is very small, the weight for time differences becomes excessively large, causing even minor time differences to be overstated and all negative samples to be treated equally. This leads to the model ignoring time-to-event information and behaving like standard contrastive learning. Conversely, with a very large $\sigma$, the weight for time differences becomes negligible, making the model insensitive to these differences. As a result, all samples receive similar weights, which also reduces the framework to standard contrastive learning.
> Our experiments ensure these findings, demonstrating that $\sigma$ must be chosen carefully. We observed that for contrastive learning to function effectively, $\sigma$ needs to be in a range where the scale of the weight applied to time differences does not excessively diverge from the model output. If the weight values become too extreme, either too large or too small, the model's performance is adversely affected. Thus, selecting an appropriate $\sigma$ that balances the impact of time differences is crucial for optimal contrastive learning performance.
> ## (Implementation Details)
> Min-max normalization was performed before the data was split into train, test, and validation sets.
> ## (Clarification on Limitations)
> We thank the reviewer for pointing out the need for clarification on limitations. We understand the importance of discussing the potential biases in detail and have incorporated your suggestions to make our proposals clearer in the updated manuscript.
>
> ---
> ### *Reference*
> [R8] D. Bahri et al., “Scarf: Self-supervised contrastive learning using random feature corruption,” ICLR, 2022.
>
> [R11] Kerdabadi et al., “Contrastive learning of temporal distinctiveness for survival analysis in electronic health records,” CIKM, 2023.

---

> > ### Comment · Reviewer_VjbS · 2024-08-08
> >
> > Thanks to all authors for their rebuttal. They have clearly gone to great lengths to address the comments that all reviewers have made. I have read the reviews from each of the reviewers, and the authors' response to them, as well as the response of the authors to my own review.
> >
> > I thank the authors for their clear explanation of the temperature coefficient, censoring, and the generation of positive and negative pairs. The manuscript would be improved by the inclusion of this text.
> >
> > I also thank the authors for performing further statistical analysis, specifically by providing the p-values associated with the Welch's t-test, to assess the performance of ConSurv relative to the other baselines. Unfortunately, I remain unconvinced by the performance of ConSurv. The proposed benefit of this model is to enhance discrimination, without sacrificing calibration. However, table E shows us that ConSurv outperforms the following models on **both** discrimination (CI) and calibration (IBS):
> > - METABRIC:  DRSA, DCS and X-CAL
> > - NWTO:  DeepSurv and X-CAL
> > - GBSG: DeepHit
> > - FLCHAIN: DRSA and DCS
> > - SUPPORT: DRSA and DCS
> > - SEER: CoxPH and DeepSurv
> >
> > In short, it seems that for 36 model-dataset pairs, only 12 show evidence for the proposed property.
> >
> > I also believe the choice to perform min-max normalisation before data splitting may have inadvertently led to data leakage, and I would expect the performance reported to be an overestimate were normalisation applied using the train/validation sets only.
> >
> > My score would change if I were convinced that the statistical performance was significant, and may be changed after further discussion.

---

> ### Author Response · Authors · 2024-08-12
> **Response to Reviewer VjbS**
>
> # [ Response to Reviewer VjbS ]
> We appreciate the reviewer for the feedback and acknowledging our efforts in addressing the reviewer’s comments. Here, we would like to provide further clarification on the following points:
>
> **Normalization and Data Leakage**:
> We apologize for any confusion caused by our initial explanation of the min-max normalization process. To clarify, we performed the min-max normalization **after** splitting the dataset, **not before**. This inadvertent miscommunication may have raised concerns about potential data leakage. We assure you that our method prevents any leakage, as normalization is conducted based solely on the training and validation sets. For verification, this procedure can be confirmed in the supplementary $\texttt{dataset.py}$ file that was submitted at the initial submission deadline.
>
> **Statistical Significance of the Performance Gain**:
> We would like to start by emphasizing that our work introduces a deep survival model leveraging contrastive learning to enhance discriminative power without sacrificing calibration. Therefore, a fairer comparison showcasing our contribution would involve comparing our model with i) variations of our model trained solely with the NLL loss or with both the NLL and ranking losses, and ii) deep survival models utilizing traditional ranking loss.
> In this context, we have evaluated the statistical significance of our results based on the p-values calculated by the Welch’s t-test. As shown in Table G, ConSurv achieves superior discriminative power (while maintaining its calibration) over its variant trained solely with the NLL loss. The results are statistically significant except for the NWTCO dataset. On the other hand, our variant trained with both the NLL and ranking losses often loses its calibration power. Contrarily, when compared with the variant trained with the NLL and ranking losses, ConSurv significantly outperforms in terms of calibration power for all the evaluated datasets while providing gains in discriminative power (but not statistically significant for the NWTCO dataset) except for the SUPPORT dataset. These results are consistent with our motivation described in the Introduction.
> A similar trend can be observed when compared with traditional ranking-based deep survival models (i.e., DeepHit and DRSA). While the gain in discriminative power is sometimes not statistically significant, our method provides statistically significant improvements in terms of calibration power, as shown in Table G.
>
> *Table G. p-value for the variants of our method*
> |Model|Metric|METABRIC|NWTCO|GBSG|FLCHAIN|SUPPORT|SEER|
> |-|-|-|-|-|-|-|-|
> |**NLL**|CI|**0.000**|0.308|**0.001**|**0.000**|**0.000**|**0.008** |
> ||IBS|**0.000**|**0.001**|0.174|0.194|**0.034**|**0.000**|
> |**NLL+Rank**|CI|**0.000**|0.424|**0.000**|**0.000**|-|**0.000**|
> ||IBS|**0.000**|**0.003**|**0.000**|**0.000**|**0.000**|**0.018** |
>
> **Analysis of Time-Dependent Performance Metrics**:
> While the CI and IBS provide valuable overall assessments of a given survival model (over the entire time horizon), these metrics cannot fully capture variations in model performance across different time points. To address this, we included time-dependent performance evaluations, namely the time-dependent C-index [R1] and the time-dependent Brier score [R2], in Appendix C.2, at three different time points (i.e., 25%, 50%, and 75% percentiles of time-to-events as in [R3, R4, R5]). It is worth highlighting that utilizing these time-dependent metrics may reveal subtle differences that might be obscured when using CI and IBS alone.
> Table H shows the number of statistically significant performance improvements over each survival model across different datasets. The improvement is considered significant if ConSurv provides statistically significant gains at at least one of the three time points. Again, we use the Welch's t-test to calculate the p-values for both time-dependent discriminative and calibration performance throughout the datasets.
> ConSurv offers superior calibration performance compared to ranking-based deep survival models (DeepHit and DRSA) and outperforms calibration-focused survival models (DCS and X-CAL) in discriminative power. Overall, compared to the deep learning baselines, ConSurv achieves statistically significant improvements in at least 4 out of 6 datasets, demonstrating its robust and consistent performance gains across various scenarios.
>
> *Table H. Number of Significant Performances Across Datasets*
> |Model|C-index|Brier score|
> |-|-|-|
> |CoxPH|3|1|
> |DeepSurv|6|6|
> |DeepHit|4|6|
> |DRSA|5|6|
> |DCS|5|5|
> |X-CAL|5|5|
> |SupWcon|5|4|

---

> ### Author Response · Authors · 2024-08-12
> **Response to Reviewer VjbS**
>
> **Evaluating Calibration from Different Perspectives**:
> We further provided the comparison in the model calibration by evaluating alternative calibration metrics -- Distributional Divergence for Calibration (DDC) [R6] and D-Calibration (D-CAL) [R7] -- that assess model calibration from a slightly different perspective. These metrics provide a more nuanced perspective on calibration quality, directly evaluating the alignment of predicted survival probabilities with uniform distributions.
> For the assessment of statistical significance of the performance gain in terms of model calibration, we have additionally included the p-values for the DDC in Table I. (Please note that the D-CAL test already incorporates p-values by its definition and thus is not considered for comparing the statistical significance of the performance gains.) Table I highlights that ConSurv achieves significant calibration gain over the benchmarks -- especially the ranking-based deep survival models -- across most of the datasets.
>
> *Table I. p-value for DDC*
> |Dataset|M|N|G|F|SU|SE|
> |-|-|-|-|-|-|-|
> |CoxPH|-|**0.012**|-|**0.018**|**0.000**|**0.017**|
> |DeepSurv|**0.030**|**0.000**|-|**0.018**|**0.000**|**0.000**|
> |DeepHit|**0.000**|**0.000**|**0.000**|**0.000**|**0.000**|**0.005**|
> |DRSA|**0.013**|-|**0.000**|**0.049**|**0.000**|**0.001**|
> |DCS|0.341|-|**0.000**|0.318|**0.000**|-|
> |X-CAL|0.926|-|0.964|**0.016**|**0.002**|-|
> |SupWcon|-|**0.005**|-|**0.043**|**0.045**|**0.000**|
>
> ---
> ## *Reference*
>
> [R1] H. Uno et al., “On the C‐statistics for evaluating overall adequacy of risk prediction procedures with censored survival data,” Statistics in Medicine, 2011.
>
> [R2] E. Graf et al., “Assessment and comparison of prognostic classification schemes for survival data,” Statistics in Medicine, 1999.
>
> [R3] Z. Wang et al., "Survtrace: Transformers for survival analysis with competing events," ACM-BCM, 2022.
>
> [R4] C. Nagpal et al., "Deep survival machines: Fully parametric survival regression and representation learning for censored data with competing risks," IEEE Journal of Biomedical and Health Informatics, 2021.
>
> [R5] C. Lee et al., "Temporal quilting for survival analysis," AISTATS, 2019.
>
> [R6]  F. Kamran et al., "Estimating calibrated individualized survival curves with deep learning," AAAI, 2021.
>
> [R7] H. Haider et al., “Effective ways to build and evaluate individual survival distributions,” JMLR, 2020.

---

> > ### Comment · Reviewer_VjbS · 2024-08-13
> >
> > Thank you to the authors for taking the time to reply again and provide further discussion and clarification.
> >
> > I have no further questions. Thank you for pointing me to the relevant code - I agree that min-max normalisation has not led to data leakage. And thanks for clarifying how your results demonstrate a statistical significant improvement over current baselines. I find the arguments above convincing.

---

> > > ### Author Response · Authors · 2024-08-13
> > > **Thanks**
> > >
> > > Dear Reviewer VjbS,
> > >
> > > We are sincerely grateful for your time and energy in the review process. In light of your satisfaction with our response, we wonder whether the reviewer would kindly consider revising the rating.
> > >
> > > Thank you, Paper 15583 Authors

---

> > > > ### Comment · Reviewer_VjbS · 2024-08-13
> > > >
> > > > I have happily raised my score from a 4 to a 6 in agreement with one of the other reviewers.

---

> > > > > ### Author Response · Authors · 2024-08-13
> > > > > **Thanks**
> > > > >
> > > > > Dear Reviewer VjbS,
> > > > >
> > > > > Thank you so much for raising the score from 4 to 6. We kindly ask if you could please ensure that the updated score is reflected in the main system.
> > > > >
> > > > > We greatly appreciate your consideration.
> > > > >
> > > > > Thank you, Paper 15583 Authors

---

> ### Comment · Area_Chair_VZW8 · 2024-08-12
> **reminder from your area chair to respond to this author feedback**
>
> Hello reviewer VjbS,
>
> Thanks for already engaging in discussion with the authors on this paper. The authors have responded in detail to your most recent comments. Please try to provide a response before the end of the author/reviewer discussion period (Aug 13 11:59pm AoE).
>
> Thanks,
> Your AC

---

### Official Review · Reviewer_1ZZS · 2024-07-09

**Soundness:** 3
**Presentation:** 3
**Contribution:** 3
**Rating:** 6
**Confidence:** 4

**Summary:**

The authors study discrete-time survival analysis, proposing to train models using a loss function that combines the NLL loss and a modified NCE loss. This NCE loss is modified to take the survival outcomes (event times) into account, mitigating the effect of potential false negatives that have small event time differences.

The method is evaluated on 4 tabular clinical survival datasets, comparing against 6 baseline models and 3 method variations. The proposed method performs well compared to all baselines.

**Strengths:**

- The paper is well-written overall.

- The proposed method is described well in Section 4.

- The proposed method is conceptually quite simple and intuitive. To add a contrastive loss term, and for this use a modified contrastive loss that takes the survival outcomes into account, makes intuitive sense.

- The proposed method seems to perform well compared to baselines, both in terms of discrimination and calibration. In particular, it performs well compared to the baseline of replacing the proposed modified NCE loss term with standard NCE.

**Weaknesses:**

- The technical novelty/innovation is perhaps somewhat limited ("just" adding a contrastive loss term).

- The experimental evaluation could perhaps be more extensive, include more datasets. Would have been nice to see the method being applied also to some non-tabular dataset.




Summary:
- Somewhat limited technical novelty, but this is a well-written paper with a simple/intuitive method that seems to perform well compared to reasonable baselines. Thus, I am definitely leaning towards accept.

**Questions:**

- Could you perhaps apply the proposed method also to some non-tabular dataset?

- The visualization in Figure 2 is really neat, could you add this for the three other datasets as well (to the appendix)?




Minor things:
- 130: "by minimizing the following negative NLL loss", remove "negative"?

- Perhaps add 1-3 sentences to the start of Section 3, just to describe what is covered here (and why this is covered, why this is relevant for your method)?

- 248: "We compare our proposed method and the benchmarks with...", "benchmarks" --> "baselines"? The same also for line 251?

**Limitations:**

Yes.

---

> ### Author Rebuttal · Authors · 2024-08-07
>
> # [ Response to Reviewer 1ZZS ]
> We thank the reviewer for the positive feedback on our work. We have addressed the comments in the updated manuscript and provided a point-by-point response below.
>
> ## (Technical Novelty about ConSurv)
> ConSurv enhances the discriminative power of survival models by using a contrastive learning framework combined with a weighted sampling technique based on the similarity of event times. Instead of directly changing risk or survival predictions, this method focuses on distinguishing samples in a latent space according to their survival outcomes. It assigns weights to samples based on differences in their survival outcomes, focusing the learning process more on important samples. While Kerdabadi et al. [R9] uses a contrastive learning framework to time-to-event analysis, ConSurv is the first approach to use hard negative samples specifically to increase discriminative power for survival analysis. In addition, ConSurv achieves excellent calibration performance without using a separate loss function specifically for calibration [R6, R10]. This is achieved as the contrastive learning process inherently preserves good calibration while enhancing discriminative capabilities.
>
> ## (Challenges of applying the proposed method to non-tabular data.)
> Unfortunately, despite the potential of the contrastive learning framework utilized in our method, we were unable to find suitable non-tabular survival data for immediate application. To the best of our knowledge, the only available dataset is the METABRIC dataset, which contains whole slide images (WSIs) of breast tumors [R4, R5] with time-to-event information for each sample. However, these WSIs present a challenge: each sample contains a varying number of image segments (collected from different locations of a tumor). Addressing this would require either domain expertise to choose the correct tumor region of interest or a deep learning technique (such as multiple instance learning) to handle samples with varying numbers of image segments. As applying augmentation in this setting is not straightforward, it was beyond the scope of our current work and remains an interesting avenue for future research.
>
> ## (Additional Experiment: t-SNE visualization)
> We have included a t-SNE visualization for each of the real-world datasets we used in our experiments in the Global PDF (Figure 2). We show experimental results for FLCHAIN and SUPPORT, where performance improvements are significant. When using the NLL alone, the representations tend to cluster tightly in a narrow space, regardless of time. However, when applying SNCE (contrastive learning with a time weight) to NLL, representations of features with similar times are more likely to be positioned closer together. This shows that our model achieves superior performance.
>
>
> ## (Clarification on Notations)
> We thank the reviewer for pointing out typos and suggestions for better clarification. We have addressed those comments in the updated manuscript.
>
> ---
> ### *Reference*
> [R4] H. Xu et al., "A whole-slide foundation model for digital pathology from real-world data," Nature, 2024.
>
> [R5] P. Mobadersany et al., "Predicting cancer outcomes from histology and genomics using convolutional networks," National Academy of Sciences, 2018.
>
> [R6] F. Kamran et al., "Estimating calibrated individualized survival curves with deep learning," AAAI, 2021.
>
> [R9] Kerdabadi et al., “Contrastive learning of temporal distinctiveness for survival analysis in electronic health records,” CIKM, 2023.
>
> [R10] M. Goldstein et al., “X-CAL: explicit calibration for survival analysis,” NeurIPS, 2020.

---

> > ### Comment · Reviewer_1ZZS · 2024-08-08
> >
> > Thank you for the response.
> >
> > I have read the other reviews and all rebuttals.
> >
> > While two of the other reviewers were leaning towards reject, I think the authors responded well overall to their concerns. I don't see any clear reason to lower my score.

---

### Official Review · Reviewer_Lfcy · 2024-07-11

**Soundness:** 2
**Presentation:** 1
**Contribution:** 2
**Rating:** 4
**Confidence:** 4

**Summary:**

The paper proposes a contrastive learning loss to regularize maximum likelihood learning of discretized survival times. The main contribution of this work is the utilization of the Laplace kernel to weigh negative pairs inversely proportional to their time difference from the anchor sample, while also accounting for censoring. Specifically, negative pairs with event times close to that of the anchor sample contribute less to the contrastive loss. Experimental results on four datasets demonstrate that the proposed approach achieves a high Concordance Index (C-Index) without a significant loss in calibration when compared to baselines.

**Strengths:**

The use of contrastive learning to improve model calibration in survival analysis seems interesting. Experimental results demonstrate the efficacy of this approach in maintaining a high C-index without significant loss in calibration, a trade-off that is difficult to achieve in practice.

**Weaknesses:**

*Given that the main contribution of this work is the application of contrastive loss to survival analysis, the paper lacks complete details for reproducibility:*
- There are no details on how the model functions $ f_{\theta}, f_{\phi}$, and $g_{\psi} $ are parametrized.
- Details on the construction of positive pairs are very sparse. No motivation is provided in terms of why the SCARF approach is chosen and why it should be expected to work with survival data.
- The paper proposes the Laplace Kernel for weighing samples; it is unclear if alternative kernels were also considered.
- The notation seems sloppy; symbols ${f}$ and $Z$ are overloaded.
- I expect calibration and the C-Index should be sensitive to $\beta $; the paper does not discuss how to choose $\beta $ and what values resulted in the performance results reported. This phenomenon has been discussed in detail in [1]. I encourage the authors to include a comprehensive discussion on this trade-off.
- It is unclear how the margin parameter $\alpha$ was chosen. I encourage the authors to provide visualizations in the main paper to demonstrate sensitivity to both $\alpha$ and $\beta $.

*The proposed approach relies on discretizing event times, which is sensitive to the time-binning approach. Additionally, it seems the contrastive loss could be an indirect way to achieve consistency in event times. I encourage the authors to also benchmark with approaches that directly predict event times such as [2].*

*The four datasets used seem to be of low sample size. I encourage the authors to also consider larger datasets such as SUPPORT and SEER.*

**References**

- [1] Qi et al., "Conformalized Survival Distributions: A Generic Post-Process to Increase Calibration", ICML 2024
- [2] Chapfuwa et al., "Calibration and Uncertainty in Neural Time-to-Event Modeling", IEEE TNNLS 2020

**Questions:**

- What are the values of $\alpha$ and $\beta $ for the reported results?
- Were alternative weighting kernels considered?
- Were alternative approaches for constructing positive pairs considered?

**Limitations:**

- The paper does not discuss modeling assumptions; for example, the assumed independent censoring mechanism could be violated in practice.

---

> ### Author Rebuttal · Authors · 2024-08-07
>
> # [ Response to Reviewer Lfcy ]
> We thank the reviewer for the valuable suggestions on our work. We have addressed the reviewer’s comments in our response below.
> ## (Sensitive Analysis of the $\alpha$)
> Introducing a margin prevents a comparable pair of samples with distant (unobserved) event times from being mistakenly treated as similar samples due to small differences between the observed event time of one sample and the censored time of the other sample within that pair. To show this effect, we have conducted an experiment using additional synthetic data by adapting the time-to-event generation process in DeepHit [R2] for both event times $T_i$ and censoring times $C_i$:
>
> $T_i \sim \exp\left((10 x_{i1})^2 + 5 x_{i3}\right)$
>
> $C_i \sim \exp\left((10 x_{i2})^2 + 5 x_{i4}\right)$
>
> Here, we randomly generated 1,000 random samples, where a sample is censored when $T_i > C_i$.
>
> In Figure 1 of the Global PDF, we compare time differences based on censoring times and the (unobserved) ground-truth event times. For each event sample, we compute the time differences between its event time and: 1) the (observed) censoring time of its corresponding comparable censored sample, and 2) the (unobserved) ground-truth event time of that same censored sample. Without a margin, censored samples with censoring times close to event times may be incorrectly treated as having similar event times, leading to very low weight assignments. To avoid this, we introduce a margin to exclude such samples when computing our contrastive loss. Overall, the margin in $|\tau_i-\tau_j| \geq \alpha$ helps prevent failures in accurately distinguishing risk between comparable censored samples.
> ## (Sensitive Analysis of the $\beta$)
> *Table D. Impact of balancing coefficient $\beta$ on performance metrics.*
> |Dataset|M||||N||||G||||F||||
> |-|-|-|-|-|-|-|-|-|-|-|-|-|-|-|-|-|
> |**Methods**|**CI**|**IBS**|**DDC**|**D-CAL**|**CI**|**IBS**|**DDC**|**D-CAL**|**CI**|**IBS**  |**DDC**|**D-CAL**|**CI**|**IBS**|**DDC**|**D-CAL**|
> |$\beta$=0.01|0.673|0.089|0.148|*0.254*|0.747|**0.069**|0.493|0.043|0.655|0.149|0.150|0.015|0.746|**0.063**|0.493|*0.353*|
> |$\beta$=0.1|0.682|0.124|0.109|*0.320*|0.759|0.091|0.470|0.095|0.677|0.171|0.156|0.012|0.775|0.108|0.364|*0.543*|
> |$\beta$=1.0|**0.706**|0.086|**0.093**|*0.524*|**0.773**|0.078|0.222|*0.428*|**0.703**|**0.138**|0.180|*0.075*|**0.816**|0.096|**0.338**|*0.465*|
> |$\beta$=10.0|0.652|0.166|0.096|*0.405*|0.759|0.093|0.344|*0.397*|0.686|0.169|**0.139**|0.002|0.777|0.107|0.354|*0.537*|
> |$\beta$=100.0|0.687|**0.078**|0.127|*0.492*|0.743|0.071|**0.463**|*0.352*|0.694|0.170|0.363|*0.057*|0.743|0.071|0.551|*0.057*|
>
> F. Kamran et al. [R6] demonstrates that while ranking loss functions used in survival analysis focus on discriminative power, they often perform poorly on calibration metrics. Similarly, Qi et al. [R7] points out the independence of discrimination and calibration. They show that models with perfect discrimination can still have significant calibration problems. This suggests that finding a good balance is crucial for improving the model performance.
>
> We investigated the relationship between discrimination and calibration, focusing on the impact of the balancing coefficient $\beta$. In Table D, additional experiments with $\beta=0.1$ and $\beta=10.0$ show that a very small $\beta$ (0.01) primarily focuses on NLL, resulting in good calibration. As $\beta$ increases, discriminative power improves while maintaining calibration performance. However, if $\beta$ is too large (beyond 1.0), it results in deviating from the concept of employing auxiliary loss, thereby leading to suboptimal discriminative and calibration performance. An optimal balance is achieved at $\beta=1.0$, performing well on both metrics. This highlights the need for hyperparameter searches to achieve optimal performance, as detailed in Appendix C.3.
>
> ## (Details of Generating Positive/Negative Samples)
> We construct positive and negative samples using the contrastive learning framework for survival data [R8]. SCARF addresses the challenge of tabular data augmentation while maintaining semantic integrity. Augmented versions $\tilde{x}^{(i)}$ of data points $x^{(i)}$ are created by selectively corrupting feature subsets using their marginal distributions. The augmented reference sample serves as the positive pair, while augmented versions from other mini-batch samples are the negative pairs. This permutation method avoids out-of-distribution issues common with noise injection described in Appendix B and is applicable to various data types. Further details have been provided in the updated manuscript.
>
> ## (Laplacian Kernel)
> We compared several kernel functions, especially the exponential and standard kernels. These kernels assign lower values to closer samples and higher values to farther samples. Our main goal, however, was to identify a kernel that not only maintains this distance-weighting property but also stabilizes at a finite value instead of increasing infinitely. Ultimately, we selected a modified version of the Laplacian Kernel, which has the property of having weights converge rather than diverge as the distance between samples increases.
> ## (Clarification on Notations)
> Each model function—$f$ (encoder $f_\theta$, hazard $f_\phi$) and $g$ (projection head $g_\psi$)—uses an MLP. Parameters were optimized using random search as detailed in Appendix E.2. We have maintained the representation as $z$ and updated the normalizing constant to $C$.
>
> ---
> ### *Reference*
> [R2] C. Lee et al., "DeepHit: A Deep Learning Approach to Survival Analysis with Competing Risks," AAAI, 2018.
>
> [R6] F. Kamran et al., "Estimating calibrated individualized survival curves with deep learning," AAAI, 2021.
>
> [R7] Qi et al., "Conformalized Survival Distributions: A Generic Post-Process to Increase Calibration", ICML 2024.
>
> [R8] D. Bahri et al., “Scarf: Self-supervised contrastive learning using random feature corruption,” ICLR, 2022.

---

> > ### Comment · Reviewer_Lfcy · 2024-08-12
> > **Official Response by Reviewer Lfcy**
> >
> > Thank you for your response and for providing additional experimental results on DRAFT, large datasets, and sensitivity analysis.
> >
> > - Unfortunately, I don't think the comparison to DRAFT is adequate; a fairer comparison would be against DATE, since Chapfuwa et al. have already demonstrated that DATE is superior to DRAFT.
> > - The connection of contrastive learning to model calibration appears to be superficial. The paper does not provide theoretical guarantees, only experimental results. Given that the experimental results highlight that $\beta$ controls the calibration-discrimination trade-off, this is a key weakness of the work. This weakness is not shared by the conformalized post-processing approach detailed by Qi et al. Hence, revisions should include comparisons to CSD and the Kaplan-Meier estimator.
> > - The paper will require significant revisions to address the sloppy mathematical notations and provide complete details on the modeling choices.
> >
> > Considering these issues, as well as the other reviews, I am inclined to maintain my score.

---

> ### Author Response · Authors · 2024-08-12
> **Response to Reviewer Lfcy**
>
> # [ Response to Reviewer Lfcy ]
>
> Once again, we would like to thank you for your invaluable feedback! We were wondering whether our response from  Aug 7 has sufficiently addressed your comment. If you have any remaining comments, please let us know. We would be happy to do our utmost to address them!

---

> ### Comment · Area_Chair_VZW8 · 2024-08-12
> **reminder from your area chair to respond to this author feedback**
>
> Hello reviewer Lfcy,
>
> The author/reviewer discussion period ends very soon (Aug 13 11:59pm AoE). It would be great if you could respond to the authors as they've taken the time to respond to your review.
>
> Thanks,
> Your AC

---

> ### Author Response · Authors · 2024-08-13
> **Response to Reviewer Lfcy**
>
> # [ Response to Reviewer Lfcy ]
>
> **Regarding the Comparisons with DRAFT and DATE**:
>
> We appreciate the reviewer's insightful comments regarding the comparison of ConSurv with DRAFT and DATE.
>
> We opted to include DRAFT, not DATE, in our comparative analysis due to the following considerations: DRAFT posits a log-normal distribution for the underlying time-to-event process, $p(t∣x)$. This allows for a direct calculation of the survival function, $S(t∣x)$, facilitating a clear and equitable comparison using the performance metrics presented in our study.
>
> Conversely, while DATE represents a significant advancement in deep survival models by directly generating time-to-event outcomes, it employs a generator that learns the underlying distribution, $p(t∣x)$, implicitly. Consequently, computing $S(t∣x)$ necessitates sampling a substantial number of samples, complicating the comparison against the reported performance metrics.
>
> Therefore, we restricted our comparison to DRAFT as a deep learning counterpart that explicitly focuses on AFT models.
>
>
> **Regarding the comparison with CSD (Qi et al.’s Method)**:
>
> We appreciate the suggestion to compare ConSurv with CSD (Qi et al.'s post-hoc calibration method), but we believe a direct comparison is not within the scope of our work. Our focus lies in improving discriminative power while preserving calibration, a distinct advancement from traditional ranking losses. CSD addresses a post-hoc approach to improve calibration of survival models, making a direct comparison less relevant.
>
> Furthermore, Qi et al.'s work was published after our submission deadline and thus couldn't be included in our original analysis. However, we acknowledge its value and will incorporate a comparison in the appendix of our final version by applying their post-hoc calibration to existing ranking-based models.
>
>
> **The Connection Between Contrastive Learning and Model Calibration**:
>
> As the reviewer pointed out,  Kamran et al. and Qi et al. have discussed the trade-off between discriminative power and calibration, where the trade-off arises because discrimination is typically assessed at an individual level, whereas calibration is evaluated at the population level. However, the parameter, $/lambda$, in the loss term proposed by Kamran et al. does not directly explain this trade-off. Instead, it controls the balance between two losses in the composite loss function, where each loss influences both the discrimination and calibration of the survival model.
>
>
> To the best of our knowledge, there is no theoretical proof on why utilizing the ranking loss may lead to poor calibration. We hypothesize that directly modifying the output of survival models (whether the hazard function in DRSA or the PMF of event times in DeepHit) can bias the network towards optimizing the approximated C-index, potentially sacrificing accurate modeling of event time distributions, as conjectured by Kamran et al.
>
> In contrast, our approach employs a contrastive learning framework that adjusts latent representations based on the similarity in the event times, implicitly promoting discriminative power without directly impacting the distribution of event times. Consequently, we were not able to observe a clear trade-off between the two loss functions, a phenomenon more commonly seen with traditional ranking losses.
>
>
> **Clarification on Notations**:
>
> We've taken steps to improve clarity by clarifying notations and providing comprehensive implementation details in both the appendix and this rebuttal. If you have specific areas that you feel still require attention, please point them out and we'll be happy to address them further.

---

### Official Review · Reviewer_RNpF · 2024-07-13

**Soundness:** 4
**Presentation:** 4
**Contribution:** 4
**Rating:** 8
**Confidence:** 4

**Summary:**

This paper presents an approach to handle survival analysis datasets using a method to improve both calibration and discrimination. This adds a bunch of novelties to the field like handling right-censoring and an SNCE loss to handle calibration and ranking simultaneously (NLL + contrastive loss). This paper presents a reliable way to look at survival analysis, write a hazard function which is quite reasonable and also present a ranking loss to take care of the domain-specific problem of stratification by risk-levels using generated patient rank list.

**Strengths:**

-> novelties added quite a lot: SNCE loss and handling right-censoring
-> ranking loss introduced can be generally used for any method aiming to generate or classify along ranked lists
-> well presented math for different loss functions and easy to follow

**Weaknesses:**

-> lack of using unstructured data
-> more clarity of censored data

**Questions:**

I understood why there is censored data, but if the authors could explain with an example of a time series what censored data looks like that would make it easier.

**Limitations:**

maybe added unstructured data like doctors notes

---

> ### Author Rebuttal · Authors · 2024-08-07
>
> # [ Response to Reviewer RNpF ]
> We thank the reviewer for the positive feedback on our work. We have included the details of censoring in the General Response Section.
> ## (Missing Experiments on Unstructured Data)
> Unfortunately, despite the potential of the contrastive learning framework utilized in our method, we were unable to find suitable unstructured survival data for immediate application. To the best of our knowledge, the only available dataset is the METABRIC dataset, which contains whole slide images (WSIs) of breast tumors [R4, R5] with time-to-event information for each sample. However, these WSIs present a challenge: each sample contains a varying number of image segments (collected from different locations of a tumor). Addressing this would require either domain expertise to choose the correct tumor region of interest or a deep learning technique (such as multiple instance learning) to handle samples with varying numbers of image segments. As applying augmentation in this setting is not straightforward, it was beyond the scope of our current work and remains an interesting avenue for future research.
>
> ---
> ### *Reference*
> [R4] H. Xu et al., "A whole-slide foundation model for digital pathology from real-world data," Nature, 2024.
>
> [R5] P. Mobadersany et al., "Predicting cancer outcomes from histology and genomics using convolutional networks," National Academy of Sciences, 2018.

---

> > ### Comment · Reviewer_RNpF · 2024-08-11
> > **Response to rebuttal**
> >
> > Thanks for the rebuttal. Agree with the future work.

---

### Author Rebuttal · Authors · 2024-08-07

# [ General Response to the Reviewers ]
We thank the reviewers for taking their valuable time to provide insightful comments and suggestions for the paper. We believe the thoughtful reviews and recommendations have substantially improved the quality of the paper. In this response, we aim to address the common comments raised by the reviewers and outline the major changes made:
- Results on additional baselines including DRAFT [R1]
- Results on additional large real-world datasets (i.e., SUPPORT and SEER)
- Explanations on censoring in time-to-event analysis

## (Additional Experiment: A New Baseline)
We have included a new baseline, DRAFT [R1], which is a deep learning approach to accelerated failure time model assuming a log-normal distribution for the underlying distribution of event times. This baseline directly models event times and hence does not rely on discretization that many deep survival models (e.g. DeepHit [R2], DRSA [R3], and our method) use.  In Table A, we show that while DRAFT demonstrates comparable performance to other deep learning baselines, our method achieves superior performance in both discrimination and calibration.

*Table A. Discrimination and calibration of survival models for DRAFT.*
| **Data**   | **Model** | **CI**   | **IBS**   | **DDC**   | **D-Cal** |
|-|-|-|-|-|-|
| **METABRIC**   | DRAFT |0.651±0.005 | 0.209±0.023 | 0.176±0.027 | 0.194±0.387 |
| 	              | ConSurv | 0.661±0.040 | 0.189±0.019 | 0.010±0.026 | 0.234±0.234 |
|  **NWTCO** | DRAFT | 0.701±0.002 | 0.125±0.009 | 0.364±0.061 | 0.000±0.000 |
|                      | ConSurv | 0.731±0.037 | 0.100±0.017 | 0.195±0.042 | 0.653±0.454 |
|  **GBSG** | DRAFT | 0.678±0.018 | 0.228±0.006 | 0.356±0.049 | 0.000±0.000 |
|                   |  ConSurv | 0.683±0.026 | 0.178±0.011 | 0.172±0.019 | 0.012±0.027 |
| **FLCHAIN** | DRAFT | 0.788±0.015 | 0.117±0.017 | 0.660±0.018 | 0.000±0.000 |
|                       |  ConSurv | 0.794±0.023 | 0.087±0.045 | 0.293±0.039 | 0.317±0.338 |


## (Additional Experiments: Large Real-world Dataset)
We thank the reviewer for suggesting additional experiments on large real-world datasets. Additionally, we have conducted experiments on the SUPPORT (8873 samples) and the SEER Prostate (54544 samples). As presented in Tables B and C, our proposed method not only significantly outperforms all baseline models in terms of discriminative power, but also demonstrates superior calibration performance across these relatively large datasets.

*Table B. Performance for SUPPORT dataset.*
| **Data**   | **Model**   | **CI**   | **IBS**   | **DDC**   | **D-Cal** |
|-|-|-|-|-|-|
| **SUPPORT**| **CoxPH**   | 0.603±0.006 | 0.196±0.006 | 0.258±0.006 | 0.000±0.000 |
|                        | **DeepSurv**| 0.596±0.014 | 0.198±0.009 | 0.244±0.035  | 0.000±0.000 |
|                        | **DRAFT**| 0.593±0.020 | 0.244±0.024 | 0.377±0.033 | 0.000±0.000
|                        | **DeepHit** | 0.615±0.007 | 0.275±0.002 | 0.336±0.003  | 0.000±0.000 |
|                        | **DRSA**    | 0.521±0.048 | 0.268±0.022 | 0.523±0.047 | 0.000±0.000 |
|                        | **DCS**     | 0.582±0.045 | 0.216±0.016 | 0.142±0.033  | 0.000±0.000 |
|                        | **X-Cal**   | 0.611±0.007 | 0.212±0.016 | 0.191±0.034 | 0.000±0.000 |
|                        | **ConSurv** | 0.615±0.007 | 0.194±0.006 | 0.142±0.014 | 0.000±0.000|

*Table C. Performance for SEER dataset.*
| **Data**   | **Model**   | **CI**   | **IBS**   | **DDC**   | **D-Cal** |
|-|-|-|-|-|-|
| **SEER** | **CoxPH**   | 0.858±0.016 | 0.009±0.001 | 0.967±0.005 | 1.000±0.000 |
|                  | **DeepSurv**| 0.764±0.037 | 0.014±0.004 | 1.000±0.000  | 1.000±0.000 |
|                  | **DRAFT**| 0.825±0.049| 0.014±0.004 | 0.804±0.017 | 0.126±0.182 |
|                  | **DeepHit** | 0.870±0.014 | 0.100±0.001 | 0.336±0.003  | 0.000±0.000 |
|                  | **DRSA**    | 0.840±0.051 | 0.017±0.001 | 0.523±0.047 | 0.000±0.000 |
|                  | **DCS**     | 0.860±0.014 | 0.010±0.001 | 0.142±0.033  | 0.439±0.513 |
|                  | **X-Cal**   | 0.838±0.041 | 0.009±0.001 | 0.191±0.034 | 0.459±0.425 |
|                  | **ConSurv** | 0.865±0.014 | 0.004±0.003 | 0.142±0.014 | 1.000±0.000|

## (Details of Censoring)
Censoring is a crucial concept in survival analysis (also known as time-to-event analysis). It refers to cases where the event of interest (e.g., death) is not observed for some individuals by the end of the study, either due to the study ending before the event occurs or the individual being lost to follow-up. Since not all events are observed, survival data are frequently right-censored; dealing with censored samples is a critical aspect in survival analysis.

Hence, the survival data for an individual patient $i$ consists of either a time-to-event $T_i \in \mathbb{R}^+$ or a time-to-censoring $C_i \in \mathbb{R}^+$, and an indicator $\Delta_i = \mathbb{I}(T_i < C_i)$; $\Delta_i = 1$ if the patient experienced the event of interest and $\Delta_i = 0$ if the patient was right-censored. Here, in survival analysis, censoring provides the information that the patient had not experienced the event (e.g., was alive) up to time $C_i$.
We will provide a clear explanation of censoring in survival analysis in the updated manuscript for a general audience.

---
### *Reference*
[R1] P. Chapfuwa et al., "Calibration and Uncertainty in Neural Time-to-Event Modeling," IEEE TNNLS, 2020.

[R2] C. Lee et al., "DeepHit: A Deep Learning Approach to Survival Analysis with Competing Risks," AAAI, 2018.

[R3] K. Ren et al., “Deep recurrent survival analysis,” AAAI, 2019.

---

### Decision · Program_Chairs · 2024-09-25

**Decision:**

Accept (poster)

**Comment:**

I would like to thank the authors and reviewers for engaging in a fruitful discussion. The reviewers mostly leaned toward acceptance (1 strong accept, 2 weak accepts, 1 borderline reject). I would consider this paper borderline, especially as the strong accept review was unusually terse, and the borderline reject review asks for additional baselines, which while reasonable to request, I realize can be challenging for the authors to fully address within the author/reviewer discussion period. I have decided to recommend acceptance for this paper, as I view the contribution of this paper as useful for the community to know even if there are very clear limitations (that I think the authors should address either with additional experiments or at least with some discussion in the manuscript text).

Specifically regarding the additional baselines requested by reviewer Lfcy: I agree with the authors that CSD was published so recently that it is fine not to compare against it, and that, furthermore, CSD is quite different in that it can be used to modify different existing survival models. However, I agree with reviewer Lfcy that comparison with DATE is actually possible (one can compute standard survival analysis metrics using DATE), so adding DATE as a baseline would be a helpful comparison.

I would encourage the authors to acknowledge that DATE and CSD exist and could potentially be good at both discrimination and calibration. Even if the authors do not plan on adding experiments, at least discussing these other methods as worthwhile to compare against in the future would be helpful. Specifically for CSD, since it works quite differently, an interesting question would be how much of a performance boost does CSD applied to various models (including the proposed method) yield (with the idea that if the base model being modified already does a great job with calibration and discrimination, then you'd expect applying CSD to be unnecessary/to lead to less of an improvement).